# Pivot Hamiltonians as generators of symmetry and entanglement

Nathanan Tantivasadakarn[1,2], Ryan Thorngren[3,2,4,5], Ashvin Vishwanath[2] and Ruben Verresen[2]

**1** Walter Burke Institute for Theoretical Physics and Department of Physics, California Institute of Technology, Pasadena, CA 91125, USA
**2** Department of Physics, Harvard University, Cambridge, MA 02138, USA
**3** Kavli Institute of Theoretical Physics, University of California, Santa Barbara, California 93106, USA
**4** Center of Mathematical Sciences and Applications, Harvard University, Cambridge, MA 02138, USA
**5** Department of Physics, Massachusetts Institute of Technology, Cambridge, MA 02139, USA

October 15, 2022

## Abstract

**It is well-known that symmetry-protected topological (SPT) phases can be obtained from the trivial phase by an entangler, a finite-depth unitary operator $U$. Here, we consider obtaining the entangler from a local 'pivot' Hamiltonian $H_{\mathbf{pivot}}$ such that $U = e^{i\pi H_{\mathbf{pivot}}}$. This perspective of Hamiltonians pivoting between the trivial and SPT phase opens up two new directions: (i) Since SPT Hamiltonians and entanglers are now on the same footing, can we iterate this process to create other interesting states? (ii) Since entanglers are known to arise as discrete symmetries at SPT transitions, under what conditions can this be enhanced to $U(1)$ pivot symmetry generated by $H_{\mathbf{pivot}}$? In this work we explore both of these questions. With regard to the first, we give examples of a rich web of dualities obtained by iteratively using an SPT model as a pivot to generate the next one. For the second question, we derive a simple criterion for when the direct interpolation between the trivial and SPT Hamiltonian has a $U(1)$ pivot symmetry. We illustrate this in a variety of examples, assuming various forms for $H_{\mathbf{pivot}}$, including the Ising chain, and the toric code Hamiltonian. A remarkable property of such a $U(1)$ pivot symmetry is that it shares a mutual anomaly with the symmetry protecting the nearby SPT phase. We discuss how such anomalous and non-onsite $U(1)$ symmetries explain the exotic phase diagrams that can appear, including an SPT multicritical point where the gapless ground state is given by the fixed-point toric code state.**

# 1  Introduction

Symmetry protected topological phases are gapped quantum phases of matter, that have been extensively studied in recent decades [1–10]. In particular, symmetry protected topological phases of bosons or spins represent a category of strongly interacting quantum matter that are nevertheless amenable to significant theoretical understanding. One way to describe SPT phases is via SPT entanglers, which are finite-depth unitary operators $U$ generating an SPT state by acting on a trivial ground state, or equivalently, an SPT model by conjugating a trivial paramagnet $H_0$, i.e., $H_{\mathrm{SPT}} = U H_0 U^\dagger$ [11–24].

Here we will study SPT entanglers which are naturally generated by Hamiltonians, $U = e^{-i\pi H_{\mathrm{pivot}}}$. We call the generating model a 'pivot Hamiltonian', as it will have the property that after a $2\pi$ rotation, it will leave the initial state or Hamiltonian invariant; in some sense we are thus pivoting around $H_{\mathrm{pivot}}$ in the space of Hamiltonians. Interestingly, there is a known example with a particularly nice pivot Hamiltonian: let us rechristen the following Ising chain as a pivot

$$H_{\mathrm{pivot}} = \frac{1}{4}\sum_n (-1)^n Z_n Z_{n+1}. \tag{1}$$

Here we have included the prefactor and sign alternation[1] for later notational convenience, and $X, Y, Z$ denote the Pauli matrices. Remarkably, a $\pi$-rotation with this Ising Hamiltonian transforms a trivial paramagnet $H_0 = -\sum_n X_n$ into the so-called cluster model:

$$H_{\mathrm{SPT}} := e^{-i\pi H_{\mathrm{pivot}}} H_0 e^{i\pi H_{\mathrm{pivot}}} = -\sum_n Z_{n-1} X_n Z_{n+1}. \tag{2}$$

The ground state of this model—called the cluster state—is known to exhibit SPT order [15, 25–27]. The fact that the Ising Hamiltonian can generate the cluster state was first pointed out in Ref. [28], albeit for a slightly different Hamiltonian including single-site terms. Here we are motivated to explore the generality of this phenomenon. In particular, given that SPT entanglers can be generated by Hamiltonians, can we obtain interesting new physics if we use, say, the cluster model (2) itself as a new pivot? We will indeed find that a non-trivial set of models can be generated in this way.

Thus far we have discussed the perspective of pivot Hamiltonians as generators of (SPT) entanglement. The second natural role that they can play is that of symmetry generators. To appreciate this, let us first point out that the interpolation $H_0 + H_{\mathrm{SPT}}$ will automatically have a $\mathbb{Z}_2$ symmetry generated by $U$ (in the assumption that our SPT phase is of order two, such that $U^2 = 1$, as is the case for the cluster phase above and all other SPT phases considered in the present work). However, given that the unitary $U$ is one element of a whole $U(1)$ group, it is natural to ask if and when $\mathbb{Z}_2$ is enhanced to a full $U(1)$ symmetry. In fact, this is the case for the above Ising example, where a straightforward calculation shows that $[H_0 + H_{\mathrm{SPT}}, H_{\mathrm{pivot}}] = 0$, (see Appendix A.1) which can be interpreted as a conservation of domain walls [12].

Hence, in addition to searching for 'nice' pivot Hamiltonians (like the Ising chain above), we ask in what cases does this pivot generate a $U(1)$ symmetry at the halfway interpolation between these two models? We report a variety of higher-dimensional generalizations, where three-site Ising and toric code pivots generate SPT phases. Moreover, we find a general criterion for when the halfway point $H_0 + H_{\mathrm{SPT}}$ has a $U(1)$ pivot symmetry. As we will discuss, such a symmetry has a mutual anomaly with the symmetry protecting the SPT phase, necessitating the $U(1)$ pivot to be a non-onsite operator. This thus gives a natural setting to expect such nonlocal $U(1)$ symmetries. There is considerable interest in the study of SPT transitions [5, 10, 25, 27, 29–49], and we argue that knowing the presence of such an unusual symmetry can be key to elucidating the structure of the surrounding phase diagram, as we will demonstrate in a variety of cases.

---

[1]Note that the alternating sign $(-1)^n$ can be eliminated by conjugating with $\prod_n X_{4n} X_{4n+1}$.

## 2 Overview

### 2.1 The general idea of pivoting

The idea of pivoting is to start with two Hamiltonians: $H_0$ with some symmetry group $G$ (typically representing a trivial paramagnet) and $H_{\text{pivot}}$ which could have less symmetry[2]. The latter will be used to evolve $H_0$ into a new Hamiltonian:

$$H(\theta) = e^{-i\theta H_{\text{pivot}}} H_0 e^{i\theta H_{\text{pivot}}}. \tag{3}$$

What makes pivoting interesting, is that $H_{\text{pivot}}$ will be chosen to function as an SPT-entangler. More precisely, we will require the following two properties:

1. $H_{\text{SPT}} := H(\pi)$ is a non-trivial SPT phase protected by $G$. (Note that this implies that $H(\theta)$ cannot be $G$-symmetric for all $\theta$.)

2. $H(2\pi) = H(0)$, i.e., there is a normalization of $H_{\text{pivot}}$ such that conjugating by a $2\pi$-rotation leaves $H_0$ invariant.

Let us note that it might be tempting to replace the second property by the slightly stronger condition that $e^{-2\pi i H_{\text{pivot}}}$ is the identity operator (which surely implies $H(2\pi) = H(0)$). We will see that in many examples, this indeed holds. However, in Sec. 3.3 we will see cases[3] where $e^{-2\pi i H_{\text{pivot}}}$ instead equals a symmetry operator (i.e., it commutes with $H_0$ and thus indeed implies $H(2\pi) = H(0)$).

In this work we will demonstrate two applications of this pivoting process. The first is that it gives a new way of building models with interesting interrelations. In particular, we already outlined above how starting from two Hamiltonians $H_0$ and $H_{\text{pivot}}$ one can construct a novel model, $H_{\text{SPT}}$. This process can be continued, e.g. using $H_{\text{SPT}}$ as a new pivot to rotate $H_0$ or $H_{\text{pivot}}$ into new models[4]. In Sec. 3, we will showcase such an example where the pivoting process gives rise to a whole series of models in distinct SPT phases, exhibiting a web of dualities.

The second application of pivoting is at SPT transitions, where the pivot can be become a symmetry. Indeed, it is instructive to consider the 1-parameter family of Hamiltonians

$$(1 - \alpha)H_0 + \alpha H_{\text{SPT}}. \tag{4}$$

By construction, the entangler $U = e^{-i\pi H_{\text{pivot}}}$ generates a $\mathbb{Z}_2$ duality $\alpha \to 1 - \alpha$. Furthermore, this duality becomes an exact $\mathbb{Z}_2$ symmetry at the midpoint ($\alpha = 1/2$), i.e., $[H_0 + H_{\text{SPT}}, U] = 0$. It has been argued before that direct SPT transitions will exhibit (either exactly or emergently) such a $\mathbb{Z}_2$ symmetry interchanging the nearby SPT phases [39, 51]; under additional conditions (see below) this has a mixed anomaly with the protecting symmetry group $G$, prohibiting a symmetric gapped phase at $\alpha = 1/2$. In our set-up, the $\mathbb{Z}_2$ unitary $U$ is generated by $H_{\text{pivot}}$, raising the question of whether the following stronger condition holds:

$$[H_0 + H_{\text{SPT}}, H_{\text{pivot}}] = 0. \tag{5}$$

---

[2]Though if $G$ is unitary, the pivot Hamiltonian has to have a strictly smaller symmetry.

[3]The same example also has the interesting property that $e^{-i\pi H_{\text{pivot}}}$ does not commute with the protecting symmetry, despite mapping symmetric models to symmetric models.

[4]For this to give rise to interesting models, one would enforce certain algebraic properties on $H_0$; moreover, using $H_{\text{SPT}}$ as a new pivot only makes sense for SPT phases protected by anti-unitary symmetries. See Sec. 3 for details.

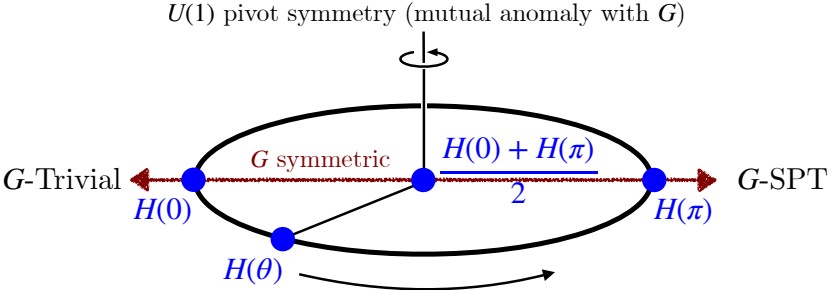

Figure 1: Visualization of the pivot process. By evolving with some pivot Hamiltonian, we create a 1-parameter family of Hamiltonians $H(\theta) = e^{-i\theta H_{\text{pivot}}} H_0 e^{i\theta H_{\text{pivot}}}$ which is $2\pi$-periodic (see Eq. (3)). Only $H(0)$ and $H(\pi)$ are symmetric with respect to some symmetry group $G$, such that these privileged points can correspond to distinct SPT phases; the other points on the circle as sketched by the compass are outside of $G$-symmetric model space. The $U(1)$ group generated by the pivot shares a mutual anomaly with $G$. In this work, we primarily focus on cases where the interpolation $\frac{1}{2}(H(0) + H(\pi))$ is invariant under this anomalous continuous symmetry group $U(1) \rtimes G$. (In a companion work [50] we present a general symmetrization procedure for creating models with this anomalous symmetry.)

Indeed, we will study a wide variety of cases where the pivot Hamiltonian generates a $U(1)$ symmetry at the SPT transition. We say that such the model has a $U(1)$ *pivot symmetry*. Moreover, we also find an instance where—even though Eq. (5) is not satisfied—the RG fixed point describing the critical point exhibits an emergent $U(1)$ pivot symmetry. This sheds new light on the nature of SPT transitions, and also serves as a guide for constructing lattice models with stable direct SPT transitions.

### 2.1.1 Anomalies

The $\mathbb{Z}_2$ entangler—and hence also the $U(1)$ pivot symmetry—often shares an anomaly with $G$, which can be taken to mean that there is no a gapped phase symmetric under both[5]. For instance, this has been proven in the case of order-two SPT phases which are not the square of another SPT phase [51]. An important practical consequence is that the full symmetry group cannot be local. More precisely, since the symmetry $G$ protecting the SPT phases is required to be on-site (for some finite unit cell), the anomaly implies that the $\mathbb{Z}_2$ entangler (or the $U(1)$ pivot) cannot be onsite[6]. Indeed, if it were onsite, it would imply the existence of a gapped symmetric phase[7], contradicting the anomaly. This thus implies that SPT transitions naturally give rise to non-local symmetries. We also note that the anomaly implies that $G$ and the pivot act as though the system lives

---

[5]It is commonly believed—although unproven—that this is equivalent to the inability to consistently gauge the symmetry; the latter is the definition of an anomaly employed in the high-energy literature.

[6]For the $\mathbb{Z}_2$ entangler, this means that it cannot be written as a tensor product of non-overlapping unitaries.

[7]This follows from the fact that a finite tensor product of a finite-dimensional representation always contains the trivial representation. To see this, it suffices to show this for the group $GL(N)$, where the complete antisymmetrization of $N$ copies gives the determinant representation, which is trivial for $GL(N)$.

on the boundary of some SPT in higher dimensions. (This SPT may be constructed by a decorated domain wall construction where entangler domain walls are decorated with the SPT that they create [52].)

Algebraically, we can describe the anomaly as follows. Suppose the $G$-SPT is classified in group cohomology [53] by the cocycle $\frac{1}{n}\omega \in H^{d+1}(G, \mathbb{R}/\mathbb{Z})$, where $d$ is the space dimension, $n$ is the order of the SPT, and $\omega$ takes values in integers mod $n$. Let $A$ be a $\mathbb{Z}_n$ gauge field (written as a 1-cocycle valued in integers mod $n$), which couples to the $\mathbb{Z}_n$ entangler (in the present work, we focus on the case $n = 2$). The anomaly may be written as $\frac{1}{n}A\omega \in H^{d+2}(\mathbb{Z}_n \times G, \mathbb{R}/\mathbb{Z})$, where we use the cup product of cocycles.

When the $\mathbb{Z}_2$ entangler is extended to a $U(1)$ pivot, the symmetry group cannot extend from $\mathbb{Z}_2 \times G$ to $U(1) \times G$. Physically this is clear, since otherwise $H(\theta)$ in Eq. (3) would be $G$-symmetric for all $\theta$; contradicting the claim that $\theta = 0, \pi$ are distinct SPT phases. Mathematically we can also see that the anomaly cannot extend to $U(1) \times G$. Indeed, if $\hat{A}$ were a $U(1)$ gauge field, $\hat{A}\omega$ would only make sense if $\omega$ can be lifted to an integer valued cocycle. However, in this case $\frac{1}{n}\omega = 0$ in $H^{d+1}(G, \mathbb{R}/\mathbb{Z})$ and so there is no entangler to begin with. For $n = 2$, we can often extend the anomaly to $U(1) \rtimes G$, where $G$ acts by automorphisms of $U(1)$. This is what happens in the examples we study. For $n > 2$, it seems the algebra of $G$ and the pivot must be larger, but we leave these cases to future work.

### 2.1.2 Pumps

In passing, we note that the anomaly implies that the pivot family $H(\theta)$ realizes a kind of generalized Thouless pump. In the language of Ref. [54], if we promote $\theta$ to a background field, we obtain a relation $\hat{A} = d\theta$, and the anomaly reduces to a topological term involving $\theta$ and the $G$ gauge field: $\theta\omega \in H_G^{d+1}(S^1, U(1))$, where we use equivariant cohomology and $G$ is understood to act on $S^1$ the same way it acts on the pivot $U(1)$. In the $\mathbb{Z}_2 \times \mathbb{Z}_2$ cluster example alluded to in the introduction (and discussed in depth in Sec. 3), one $\mathbb{Z}_2$ acts by reflecting the circle and the other protects a charge which is pumped around the cycle. At the SPT point, the edge mode may be considered as a boundary transition where two edge states of opposite charge cross. Equivalently, if we consider evolving with pivot on a manifold with a boundary, in all of these examples, we find that an SPT of one lower dimension is pumped to the boundary [55, 56].

### 2.2 Condition for enhanced $U(1)$ pivot symmetry

We have already noted that it is an interesting question to ask when Eq. (5) is satisfied. To this end, we have derived a very simple sufficient condition which ensures that for a general class of pivot Hamiltonian, the $\mathbb{Z}_2$ symmetry is enhanced to a full $U(1)$ pivot symmetry at the midpoint $H_0 + H_{\text{SPT}}$. This captures all the examples studied in this work with a purely diagonal pivot.

Let the trivial Hamiltonian be $H_0 = -\sum_v X_v$ and suppose that the pivot is a sum of local terms consisting of a product of Pauli-$Z$ operators up to a sign. Schematically,

$$H_{\text{pivot}} = \frac{1}{N} \sum \left( \pm \prod Z \right) \tag{6}$$

where $N$ is the largest integer such that $e^{-2\pi i H_{\text{pivot}}} = 1$.

Let $k$ be the number of times that a Pauli-$Z$ at a given vertex[8] appears in Eq. (6). One can show that as long as $k < N$, then $H_0 + H_{\text{SPT}}$ will commute with $H_{\text{pivot}}$, i.e., the

---

[8]If $k$ depends on the vertex, we define $k$ to the maximal value over all vertices.

$\mathbb{Z}_2$ entangler is enhanced to an exact $U(1)$ symmetry! (E.g., for the 1D pivot in Eq. (1), we have $2 = k < N = 4$, guaranteeing the $U(1)$ symmetry of $H_0 + H_{\text{SPT}}$.)

This theorem is derived in Appendix B. Since the key steps are simple to state, let us give a brief summary here. One can always define a generalized Kramers-Wannier (KW) transformation which maps the pivot to an *onsite* operator; this map has the following two properties:

1. $H_{\text{pivot}} \to \frac{1}{N} \sum Z$,

2. $H_0$ and $H_{\text{SPT}}$ map to $k$-local terms.

Note that the first property implies that the $\mathbb{Z}_2$ entangler can be written as

$$e^{-i\pi H_{\text{pivot}}} = e^{i\frac{2\pi}{N}\sum Z/2}. \tag{7}$$

In other words, in the KW dual language, $H_0 + H_{\text{SPT}}$ is guaranteed to have a $\mathbb{Z}_N$ symmetry. However, the second property—namely $k$-locality—implies that the largest charge that any local Hamiltonian term can contain is $k$. Hence, if $k < N$, the $\mathbb{Z}_N$ symmetry automatically implies $U(1)$ symmetry.

In 1D, we also discuss examples where the pivot is not purely diagonal. In those cases, we find that it is possible for the pivot to square to the symmetry of the Hamiltonian (see Sec. 3.3).

## 2.3 Summary of examples studied in this work

In most of the examples we study, we start with the trivial phase given by the Hamiltonian $H_0 = -\sum X$, whose ground state is the trivial paramagnet. Using a pivot whose ground state realizes a different phase, we create an SPT by a $\pi$ rotation. A summary of some of the pivot Hamiltonians considered in this paper—and the resulting generated SPT phases—are given in Table 1. We can choose different protecting symmetries. In the table we include both unitary and anti-unitary examples.

In Sec. 3 we discuss the simplest example, where the pivot is an Ising Hamiltonian, which we have already touched upon in the introduction. In 1D, this already reproduces a variety phases of integrable spin chains and their transitions. Moreover, we explore what happens when using SPT models as pivot Hamiltonians for generating yet more models, and we uncover a whole array of models exhibiting rich physics.

We show how this 1D example can be bootstrapped to higher dimensions in Sec. 4 by using the decorated domain wall construction to create a 2D pivot out of the 1D pivot. We obtain a three-body Ising pivot, which can be interpreted as a staggered Baxter-Wu [57] Hamiltonian, and the SPT it generates is protected by $G_U = \mathbb{Z}_2^3$ [17] or $G_A = \mathbb{Z}_2^2 \times \mathbb{Z}_2^T$. Similar to the 1D case, we find that the direct interpolation on the triangular lattice also has a $U(1)$ pivot symmetry. In a companion work, we study how this $U(1)$ pivot symmetry can be used to stabilize an exotic SPT transition described by deconfined quantum criticality [50].

Finally in Sec. 5, we use the toric code as a pivot and study phase transitions between SPT phases protected by higher-form symmetries; more physically, one can interpret these as SPT phases protected by conventional symmetries but in a constrained Hilbert space (e.g., only closed loop configurations). We find both a transition with dynamical critical exponent $z_{\text{dyn}} = 2$ as well as an $O(2)/\mathbb{Z}_2$ CFT, where the quotient denotes having gauged (orbifolded) a $\mathbb{Z}_2$ symmetry in the regular $O(2)$ criticality. The latter is example of case where lattice $\mathbb{Z}_2$ emerges to $U(1)$ pivot symmetry in the IR. Interestingly, the $z_{\text{dyn}} = 2$ multicritical point, although gapless, has the toric code wavefunction as its ground state, and in general, we also find an exactly solvable path connecting trivial and SPT passing through this multicritical point.

| Pivot | **Sec. 3**<br>**Ising in 1D** | **Sec. 4**<br>**Multi-body Ising in $d$D** | **Sec. 5**<br>**Toric code in $d$D** |
|---|---|---|---|
| Lattice | 1D | $(d+1)$-colorable | Voronoi cellulation |
| **Unitary symmetry** | $\mathbb{Z}_2^{d+1}$ | $\mathbb{Z}_2^2$ | N/A |
| **Anti-unitary symmetry** | $\mathbb{Z}_2 \times \mathbb{Z}_2^T$ | $\mathbb{Z}_2^d \times \mathbb{Z}_2^T$ | $B^{d-1}\mathbb{Z}_2 \times \mathbb{Z}_2^T$ |
| $H_0$ | $-\sum_v X_v$ | $-\sum_v X_v$ | $-\sum_{\lozenge_{d-1}} X_{\lozenge_{d-1}}$ |
| $H_{\text{pivot}}$ | $\dfrac{1}{4}\sum_n (-1)^n Z_n Z_{n+1}$ | $\dfrac{1}{2^{d+1}}\sum_{\lozenge_d}(-1)^{\lozenge_d}\prod_{v\in\lozenge_d} Z_v$ | $\dfrac{1}{4}\sum_{\lozenge_d}\prod_{\lozenge_{d-1}\in\partial(\lozenge_d)} Z_{\lozenge_{d-1}}$ |
| **Multicritical point** | BEC (Sec. 3.2.1),<br>KT (Sec. 3.2.2) | $SO(5)$ DQCP<br>for $d=2$ [50] | BEC/$\mathbb{Z}_2$ (Sec. 5.3),<br>$O(2)/\mathbb{Z}_2$ (Sec. 5.4) |

Table 1: Summary of symmetries and the corresponding pivot Hamiltonians $H_{\text{pivot}}$. Starting with the product state Hamiltonian $H_0$, the ground state of the Hamiltonian $H_{\text{SPT}} = e^{-i\pi H_{\text{pivot}}} H_0 e^{i\pi H_{\text{pivot}}}$ is an SPT protected by either unitary or anti-unitary symmetry. Furthermore, for the first and last column, and for the middle column in $d=2$ on the triangular lattice, we find that $[H_0 + H_{\text{SPT}}, H_{\text{pivot}}] = 0$, giving a $U(1)$ pivot symmetry at the midway point.

# 3  Pivoting with the Ising model: $\mathbb{Z}_2^2$ and $\mathbb{Z}_2 \times \mathbb{Z}_2^T$ SPT in 1+1D

As a first example, we will return to the Ising model as a pivot. This could be explored for lattices in any dimension. That is, it can be used to create cluster states on arbitrary lattices. However, using the general argument in Sec. 2.2, only the 1D lattice will give rise to an enlarged $U(1)$ pivot symmetry at the SPT transition (indeed, for lattices with even coordination number we find in Eq. (6) the normalization pre-factor $N = 4$, whereas the occurrence $k$ of each Pauli-$Z$ equals the coordination number). Hence, we restrict to this one-dimensional case in this section. We study the resulting 1D SPT in Sec. 3.1 and various phase transitions in Sec. 3.2 which leverage the $U(1)$ pivot symmetry. Lastly, Sec. 3.3 shows how applying the pivoting procedure several times—now using the SPT models as pivots—generates a whole class of interesting models, with various dualities interrelating them.

## 3.1  The pivot and the cluster SPT

We consider a 1D nearest-neighbor Ising Hamiltonian in Eq. (1) as defined in the introduction. The prefactor of $1/4$ is chosen such that

$$e^{-2\pi i H_{\text{pivot}}} = \prod_n (Z_n Z_{n+1}) = 1 \tag{8}$$

for periodic boundary conditions. According to the general prescription laid out in Sec. 2, we will consider the $\mathbb{Z}_2$ unitary generated by the above pivot. Rewriting the Pauli-$Z$ in

terms of the number operator $s_n = \frac{1-Z_n}{2}$, we find

$$U = e^{-\pi i H_{\text{pivot}}} = e^{-\frac{\pi i}{4} \sum_n (-1)^n (1 - 2s_n - 2s_{n+1} + 4s_n s_{n+1})}$$

$$= e^{\pi i \sum_n s_n s_{n+1}} = \prod_n CZ_{n,n+1}. \tag{9}$$

Here, $CZ$ is the controlled-$Z$ operator defined as

$$CZ_{ij} = (-1)^{s_i s_j} = \frac{1 + Z_i + Z_j - Z_i Z_j}{2}. \tag{10}$$

Starting with the product state Hamiltonian $H_0 = -\sum_n X_n$, the SPT Hamiltonian that results from the evolution using the pivot is thus indeed the cluster Hamiltonian in Eq. (2).

The 1D cluster state is a non-trivial SPT phase protected by either a $\mathbb{Z}_2^2$ [25, 26] or $\mathbb{Z}_2 \times \mathbb{Z}_2^T$ [15, 27] symmetry. In the former, the symmetry group is generated by

$$P_1 = \prod_n X_{2n+1} \qquad\qquad P_2 = \prod_n X_{2n} \tag{11}$$

and in the latter,

$$P = P_1 P_2 = \prod_n X_n \qquad\qquad \mathcal{T} = K. \tag{12}$$

Here $K$ corresponds to complex conjugation in the diagonal basis.

We first explore the topic of SPT transitions for this model. Later, in Sec. 3.3, we will study other models which arise when taking the cluster model itself as a pivot.

## 3.2   SPT transitions

To explore the transitions between the trivial and SPT phase, we first consider the direct interpolation as in Eq. (4):

$$H = (1 - \alpha)H_0 + \alpha H_{\text{SPT}} \tag{13}$$
$$= -\sum_n \left[ (1 - \alpha)X_n + \alpha Z_{n-1} X_n Z_{n+1} \right].$$

This model has been well-studied even before the notion of SPT phases arose. In particular, it is known that it can be solved in terms of free fermions by using a Jordan-Wigner transformation [58, 59] or can be mapped directly onto the XY chain via a Kramers-Wannier (KW) transformation [60]. For completeness, we review this KW mapping in Appendix A.4. In the main text, we will highlight some of the salient features of this model (and perturbations thereof) without using non-local variables, which could otherwise obscure some of the physics at play.

Firstly, at the halfway point ($\alpha = 1/2$) the model enjoys a $U(1)$ symmetry, i.e. $H_0 + H_{\text{SPT}}$ commutes with $H_{\text{pivot}}$, as was observed in [12]. This is an example of what we mentioned in Sec. 2: although $H_0 + H_{\text{SPT}} = H_0 + U H_0 U^\dagger$ by definition will commute with the $\mathbb{Z}_2$ operator $U = e^{-i\pi H_{\text{pivot}}}$, sometimes this can be enhanced to a full $U(1)$. This is indeed the case here, which follows from the general result in Sec. 2.2: we have $N = 4$ whereas $k = 2$, such that $k < N$. This $U(1)$ pivot symmetry can also be directly demonstrated as shown in Appendix A.1.

Secondly, this halfway point is critical and is described by a compact boson CFT. For completeness, we describe the lattice-continuum correspondence in Appendix A.2.

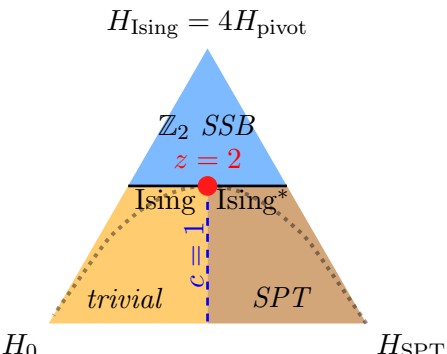

Figure 2: Phase diagram of the 1D cluster SPT transition perturbed by the Ising pivot: Hamiltonian (14) [61–63] plotted in barycentric coordinates. The central vertical axis has an explicit $U(1)$ symmetry generated by $H_{\text{pivot}}$, which rotates the trivial and SPT phases into one another. This stabilizes a compact boson transition ('$c = 1$') which eventually gaps out upon tuning the filling, making way for a symmetry-breaking phase which satisfies the mutual anomaly between the $U(1)$ symmetry and the $\mathbb{Z}_2 \times \mathbb{Z}_2^T$ symmetry protecting the SPT phase. In the center of the phase diagram, we observe a multicritical point with dynamical critical exponent $z = 2$; this can be perturbed into two topologically distinct symmetry-enriched versions of the Ising criticality [46]. The dotted line is an exactly-solvable frustration-free line which tunes through the multicritical point [61–63]. It is instructive to compare this 1D phase diagram to a 2D analogue in Fig. 6.

In preparation for our generalizations to higher dimensions, we now explore what happens upon perturbing this critical point with either the pivot (tuning the chemical potential) or a next-nearest neighbor Ising interaction (which is known to contain the marginal parameter, see Appendix A.2).

Both perturbations will give rise to an interesting multicritical point in the phase diagram. While this multicriticality might seem of secondary interest in the one-dimensional setting, it is where we will find a continuous SPT transition in the higher-dimensional examples (indeed, the extended continuous $c = 1$ criticality which is generic in the 1D setting will typically be replaced by a first order line in higher dimensions, whereas the multicriticality turns out to be more robust).

### 3.2.1 Adding the pivot: BEC multicriticality

We add the pivot Hamiltonian to Eq. (13):

$$H = (1 - \alpha)H_0 + \alpha H_{\text{SPT}} + hH_{\text{pivot}} \tag{14}$$
$$= -\sum_n \left[(1 - \alpha)X_n + \alpha Z_{n-1}X_nZ_{n+1} + h(-1)^n Z_nZ_{n+1}\right]$$

Note that since $H_{\text{pivot}}$ breaks the $\mathbb{Z}_2^2$ symmetry, the above Hamiltonian exhibits an SPT phase which is only protected by $G_A = \mathbb{Z}_2 \times \mathbb{Z}_2^T$.

This phase diagram has been studied before; see e.g. Refs. [61–63]. We reproduce it in Fig. 2. Let us first focus on the self-dual line $\alpha = 1/2$, where the Hamiltonian has an exact $U(1)$ pivot symmetry. For small $h$, the compact boson CFT is stable to perturbing with the current operator (we simply tune the filling). On the other hand, it is clear that for large $h \to \infty$, we have a gapped phase with two degenerate ground states with a four-site

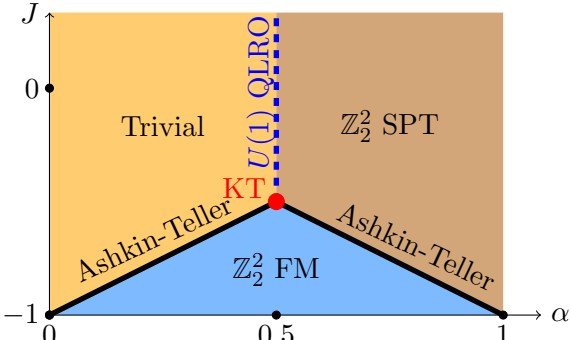

Figure 3: Phase diagram of the 1D cluster SPT transitions perturbed by a second-nearest-neighbor Ising interaction: Hamiltonian (15), which is dual to the XXZ chain by gauging the diagonal $\mathbb{Z}_2$ symmetry. The vertical $\alpha = 1/2$ has an exact $U(1)$ symmetry generated by the Ising pivot $H_{\text{pivot}}$. The blue dashed line is a compact boson transition between the trivial and cluster SPT phase. As we tune $J$ to be large and negative, there is a KT transition into a ferromagnet with fourfold ground state degeneracy. This symmetry-broken phase is separated from the symmetry-preserving phases by an Ashkin-Teller criticality.

unit cell of the pattern 0011 or 1100. These two regimes are separated by a multicritical point with dynamical critical exponent $z = 2$ [61–69], denoted by a red point in Fig. 2. This can be interpreted as a limiting case where we are still in maximal filling but there is gapless quadratic dispersion above this ground state. Unlike the standard Bose-Einstein condensate (BEC) transition, the 'empty' ground state is not completely structureless: it is given by the aforementioned twofold degenerate manifold of states. This is a consequence of the non-trivial structure of the pivot Hamiltonian, which is necessarily non-onsite due to the mutual anomaly with $\mathbb{Z}_2 \times \mathbb{Z}_2^T$ (the ground state of $H_{\text{pivot}}$ spontaneously breaks this symmetry).

The $U(1)$ pivot symmetry is explicitly broken by tuning $\alpha$ away from $1/2$. In the field theory this corresponds to perturbing the compact boson with $\cos\varphi$, which naturally flows to $\varphi = 0$ (trivial phase) or $\varphi = \pi$ (SPT phase) depending on the sign of this perturbation (see Appendix A.2). At the multicritical point, it triggers a flow to Ising criticality, as is also evidenced in Fig. 2. Due to the remaining $\mathbb{Z}_2 \times \mathbb{Z}_2^T$ symmetry, these two Ising critical lines are topologically distinct (i.e., there is no symmetric path of Ising-critical Hamiltonians connecting them); these distinct criticalities have been studied before in Ref. [46]. In particular, whether the disorder operator $\mu(x)$ is real or imaginary serves as a topological invariant distinguishing the two Ising criticalities.

### 3.2.2 Next-nearest neighbor Ising perturbation: KT multicriticality

In the previous example, we explicitly added the pivot symmetry. The downside is that this explicitly breaks part of the unitary $\mathbb{Z}_2^2$ symmetry. Sometimes it can be advantageous to preserve it, since part of it anticommutes with the pivot: this gives an additional particle-hole symmetry which can help to stabilize an interesting multicritical point. To illustrate this in a simple model, we now preserve the 'charge-conjugation' symmetry $P_1$ and add a next-nearest neighbor Ising interaction $H_{\text{NNN}} = \sum_n Z_{n-1}Z_{n+1}$ with strength

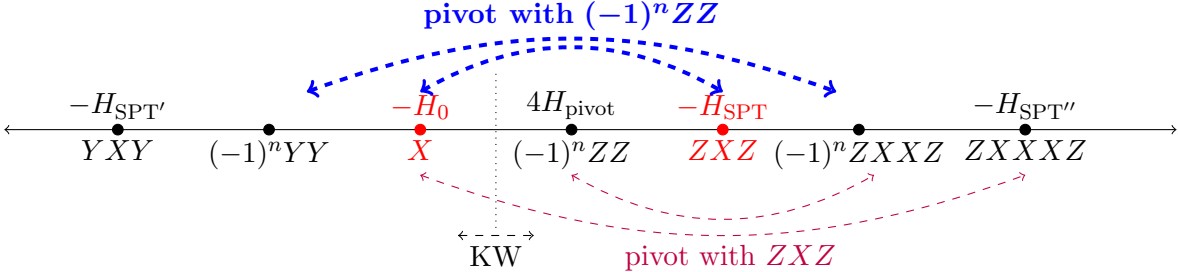

Figure 4: Using the SPT Hamiltonian as a pivot gives rise to a different SPT phase. Indeed, the cluster chain $H_{\text{SPT}} = -\sum_n Z_{n-1}X_nZ_{n+1}$ and the longer-range $H_{\text{SPT}''} = -\sum_n Z_{n-2}X_{n-1}X_nX_{n+1}Z_{n+2}$ are known to be in distinct phases protected by $\mathbb{Z}_2 \times \mathbb{Z}_2^T$ symmetry [27]. Continuing this process generates a whole one-dimensional array of models which can be rotated into one another. In particular, pivoting around a given Hamiltonian corresponds to spatial inversion in this abstract 'model space' (ilustrated by the blue and green dashed lines). non-local Kramers-Wannier duality corresponds to reflecting across an axis which is positioned *between* sites.

$J$. That is,

$$
\begin{aligned}
H &= (1-\alpha)H_0 + \alpha H_{\text{SPT}} + JH_{\text{NNN}} \\
&= \sum_n \left( -(1-\alpha)X_n - \alpha Z_{n-1}X_nZ_{n+1} + JZ_{n-1}Z_{n+1} \right).
\end{aligned}
\tag{15}
$$

Note that this model still commutes with $H_{\text{pivot}}$. The phase diagram is shown in Fig. 3 and is dual to the integrable XXZ chain (see Appendix A.4). We again find a multicritical point, which now is a full-fledged CFT rather than a $z_{\text{dyn}} = 2$ theory. In Appendix A.3, we show how it is described by Kosterlitz-Thouless (KT) criticality.

## 3.3 Web of dualities: pivoting with the pivoted

A nice application of the pivoting process is that it allows one to obtain a whole family of exactly solvable Hamiltonians starting with two Hamiltonians. For example, given that we have used the Ising Hamiltonian as a pivot, it is natural to ask what happens when we use the SPT Hamiltonian itself as a pivot. Normalization implies that we should evolve by $H_{\text{SPT}}$ by an angle of $\pi/4$ (since doing that twice commutes with $H_0$, conforming to the first property listed in Sec. 2.1), and we find that

$$
e^{-i\pi H_{\text{SPT}}/4} H_0 e^{i\pi H_{\text{SPT}}/4} = -\sum_n Z_{n-2}X_{n-1}X_nX_{n+1}Z_{n+2}
\tag{16}
$$

which is a different SPT phase protected by $\mathbb{Z}_2 \times \mathbb{Z}_2^T$ [27]. In fact, $\mathbb{Z}_2 \times \mathbb{Z}_2^T$ admits three non-trivial SPT phases which form a $\mathbb{Z}_2 \times \mathbb{Z}_2$ group. The cluster chain is one, Eq. (16) is another; the third is given by $H = -\sum_n Y_{n-1}X_nY_{n+1}$ which can be obtained by, e.g., pivoting the cluster chain around $H_0$ (effectively swapping $Y \leftrightarrow Z$).

This process can be continued to generate a whole array models. It is instructive to represent these models as living on the number line as in Fig. 4, representing the basis of our 'model space'. The action of pivoting around a given model then geometrically corresponds to 'reflecting' model space around the pivot Hamiltonian of choice. To make

this precise, let us define a family of Hamiltonians $H_k$ for $k \in \mathbb{Z}$ as

$$H_k = \begin{cases} -\sum_n (-1)^{kn} Y_n X_{n+1} \cdots X_{n+k-1} Y_{n+k}; & k < 0 \\ -\sum_n X_n; & k = 0 \\ -\sum_n (-1)^{kn} Z_n X_{n+1} \cdots X_{n+k-1} Z_{n+k}; & k > 0 \end{cases} \tag{17}$$

Then we can identify $H_0 = H_0$, $H_{\mathrm{pivot}} = \frac{1}{4} H_1$, $H_{\mathrm{SPT}} = H_2$. These Hamiltonians correspond to the different points of Fig. 4. One can straightforwardly show that pivoting $H_k$ around $H_{k_0}$ (for arbitrary $k, k_0 \in \mathbb{Z}$) is given by:

$$U_{k_0} H_k U_{k_0}^\dagger = H_{2k_0 - k} \text{ with } U_k := e^{-i\pi H_k/4}. \tag{18}$$

Note that this action $H_k \to H_{2k_0 - k}$ indeed corresponds to the geometric idea of spatially inverting the integer labels such that $k = k_0$ is invariant (i.e., 'the pivot'). Moreover, by concatenating two such inversions, one can implement arbitrary two-site translations in 'model space'. In particular, conjugating by $U_a U_0$ implements a shift $H_k \to H_{k+2a}$. More generally, by exponentiating Eq. (18), we see that $U_a U_b = U_{a+c} U_{b+c}$, as we expect for inversion.

Moreover, the Kramers-Wannier duality (see Appendix A.4) acts naturally on this family as $H_k \to H_{1-k}$. I.e., whereas pivots act as inversions around 'sites' of our model space, KW duality corresponds to inverting *between* sites, as also sketched in Fig. 4. By concatenating this with $U_0$, we can thus implement arbitrary translations in model space. Conceptually, it is interesting to note that $H_0 = -\sum_n X_n$ is sufficient to generate this whole space: a first KW transformation creates $H_1$, after which one can start pivoting to create all $H_k$.

The ground state physics of these spin chains has been discussed before in Ref. [27]. In particular, for $H_k$ with odd $k$, the ground state spontaneously breaks $\prod_n X_n$ symmetry; nevertheless, these can still form distinct phases (e.g., $H_3$ is also a non-trivial SPT phase protected by $T = K$, in addition to spontaneously breaking $\prod_n X_n$). In contrast, $H_k$ with even $k$ is symmetry-preserving. We find a similar difference in the pivots: $U_k$ for odd $k$ are all $\mathbb{Z}_2$ operators, whereas for even $k$ we find that $U_k^2 = \prod_n X_n$, making $U_k$ a $\mathbb{Z}_4$ operator. This thus gives a variety of instances where the SPT-entangler is not $\mathbb{Z}_2$ (despite creating a $\mathbb{Z}_2$ SPT phase in these cases). Relatedly, for even $k$, $U_k$ does not commute with $T = K$; instead we find

$$T U_k T = U_k^\dagger = U_k^3 = U_k \times \prod_n X_n. \tag{19}$$

Since $\prod_n X_n$ is a symmetry of all $H_k$, we still have that $U_k$ maps symmetric models to symmetric models. This gives an example where the group formed by the symmetry protecting the SPT (here $\mathbb{Z}_2 \times \mathbb{Z}_2^T$) and the SPT-entangler (here $\mathbb{Z}_4$) is not a direct product: instead we find $(\mathbb{Z}_2 \times \mathbb{Z}_4) \rtimes \mathbb{Z}_2^T$.

We point out that pivot symmetries straightforwardly generalize. We have already discussed how $H_0 + H_2$ commutes with the Ising pivot $H_1$. This generalizes to the following more general property:

$$\forall a, b \in \mathbb{Z} : [H_{a-b} + H_{a+b}, H_a] = 0. \tag{20}$$

The simplest way to prove it is by using the shift property to see that it is equivalent to the claim that $[H_{-b} + H_b, H_0] = 0$. This claim can be verified on sight, since $H_0 \propto \sum X$ is the standard on-site $U(1)$ generator and $H_{-b} + H_b$ is manifestly $U(1)$-symmetric.

Lastly, let us also remark that by a Jordan-Wigner transformation, these spin chains can be related to longer-range Kitaev chains [27].

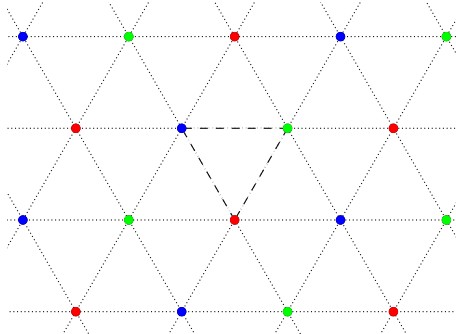

Figure 5: The triangular lattice. Qubits are placed on the vertices, which are colored in red green and blue. The $\mathbb{Z}_2^3$ symmetry is generated by spin flips on each of the individual colors (which we label, the A,B,C sublattices). The pivot is a alternate sum of a three-body Ising interaction $ZZZ$ over all triangles.

# 4   Pivoting with three-body Ising model: $\mathbb{Z}_2^3$ and $\mathbb{Z}_2^2 \times \mathbb{Z}_2^T$ SPT in 2+1D

In this section, we will demonstrate how to bootstrap ourselves up in dimensionality. In Sec. 4.1, we show that the 1D pivot in the previous section can be naturally used to construct a 2D pivot via the decorated domain wall construction [52]. In Sec. 4.2, we discuss the properties of the resulting SPT model. In Sec. 4.3, we discuss pivot symmetries on various lattices. A generalization of this construction to higher dimensions is reported in Appendix C.

## 4.1   Decorated domain wall pivots

First, we recall the decorated domain wall construction. Given a 1D $G$-SPT of order two, we can construct a 2D SPT protected by $\mathbb{Z}_2 \times G$ by condensing $\mathbb{Z}_2$ domain walls, which are attached with the 1D SPT. Operationally, we will demonstrate that using a 1D pivot, which generates a $\mathbb{Z}_2$ unitary that creates the 1D SPT, we can design a 2D pivot which performs precisely this decoration.

To set this up, we will work on a triangular lattice with qubits placed on each vertex. The lattice is three-colorable as shown in Fig. 5, and we define the subset of the three colors $A$, $B$, and $C$, respectively. We define the $\mathbb{Z}_2$ symmetry to act on only one of the sublattices, i.e., $\prod_{a \in A} X_a$. Furthermore, the remaining $G$ symmetry will act on the $B$ and $C$ sublattices. Notably, the domain wall of the $A$ sublattice forms a closed loop along the $BC$ sublattices. Therefore, we can use the 1D pivot to "decorate" the SPT whenever a domain wall of the $A$ sublattice is present. In order to be able to do this, we must be able to express the pivot Hamiltonian as a sum of local terms: $H_{\text{pivot}}^{(1)} = \sum_{\langle b,c \rangle} H_{\text{pivot},bc}^{(1)}$, where $H_{\text{pivot},bc}^{(1)}$ acts only on two sites $b$ and $c$. For example, if the pivot is the Ising Hamiltonian, then $H_{\text{pivot},bc}^{(1)} = \frac{1}{4} Z_b Z_c$.

We claim that the local terms that make up the 1D pivot Hamiltonian can be used to generate the 2D pivot Hamiltonian for the $\mathbb{Z}_2 \times G$ SPT, namely

$$H_{\text{pivot}}^{(2)} = \frac{1}{2} \sum_{\Delta_{abc}} (-1)^{\Delta_{abc}} Z_a H_{\text{pivot},bc}^{(1)} \tag{21}$$

where $\Delta_{abc}$ are triangles whose vertices are colored once by each of the $A, B, C$ sublattices, and $(-1)^{\Delta_{abc}}$ is $+1$ for all up triangles and $-1$ for all down triangles.

To see why this pivot does the job, we consider the $\mathbb{Z}_2$ unitary it generates.

$$e^{\pi i H_{\text{pivot}}^{(2)}} = \prod_{\Delta_{abc}} e^{\frac{\pi i}{2}(-1)^{\Delta_{abc}}(1-2s_a)H_{\text{pivot},bc}^{(1)}} \tag{22}$$

$$= \prod_{\Delta_{abc}} e^{\frac{\pi i}{2}(-1)^{\Delta_{abc}}H_{\text{pivot},bc}^{(1)}} \prod_{\Delta_{abc}} e^{\pi i(-1)^{\Delta_{abc}}s_i H_{\text{pivot},bc}^{(1)}} \tag{23}$$

where again we have written $Z_a = 1 - 2s_a$ in terms of the number operator $s_a$ for each site $a \in A$. First, we notice that on each edge $(bc)$, the term $\prod_{\Delta_{abc}} e^{\frac{\pi i}{2}(-1)^{\Delta_{abc}}Z_j Z_k}$ pairwise cancels exactly because of the alternating sign $(-1)^{\Delta_{abc}}$ of the two adjacent triangles. The remaining term is not affected by this sign, and therefore we conclude that

$$e^{\pi i H_{\text{pivot}}^{(2)}} = \prod_{\Delta_{abc}} \left( e^{\pi i H_{\text{pivot},bc}^{(1)}} \right)^{s_a}. \tag{24}$$

Now let us focus again at each edge $(bc)$, which is the position of a domain wall formed by the two adjacent $A$ spins, which we call $a_1$ and $a_2$. We see that, on this edge, the contribution of the unitary from the two adjacent triangles $\Delta_{a_1 bc}$ and $\Delta_{a_2 bc}$ is

$$\left( e^{\pi i H_{\text{pivot},bc}^{(1)}} \right)^{s_{a_1}+s_{a_2}}. \tag{25}$$

Thus, we see that when a domain wall is present, ( $s_{i_1} + s_{i_2} = 1 \pmod 2$), the unitary generated by a $\pi$-evolution of the 1D pivot is implemented, entangling the 1D SPT, and when the domain wall is absent ( $s_{i_1} + s_{i_2} = 0 \pmod 2$), the identity is implemented. Thus, starting with the product state in the $X$-basis, which is a superposition of all spins in the $Z$-basis. The unitary generated by the 2D pivot attaches the 1D SPT along the domain walls of the spins as desired.

## 4.2   SPT model

We now return to a concrete example, where the 1D pivot is the Ising Hamiltonian $H_{\text{pivot},bc}^{(1)} = \frac{1}{4}Z_b Z_c$[9]. That is, our the 1D $G$-SPT created from this pivot is the 1D cluster state protected by either $G = \mathbb{Z}_2^2$ or $\mathbb{Z}_2 \times \mathbb{Z}_2^T$ where the symmetries are generated by $P_B = \prod_{b \in B} X_b$ and $P_C = \prod_{c \in C} X_c$ or $P_{BC} = \prod_{v \in B,C} X_v$ and $T = K$, respectively. The 2D pivot in this case is therefore

$$H_{\text{pivot}}^{(2)} = \sum_{\Delta_{abc}} (-1)^{\Delta_{abc}} Z_a Z_b Z_c. \tag{26}$$

It is worth comparing this pivot to other three-body "Ising-type" interactions in the literature. The Hamiltonian with ferromagnetic couplings for both up and down triangles on the triangular lattice is known as the Baxter-Wu model [57] whose ground state is a ferromagnet with four-fold degeneracy. If only the down triangle terms are present, the Hamiltonian is known as the Newman-Moore model [70] which has a subextensive ground state degeneracy and fractal symmetries. In contrast, the ground state of the above pivot Hamiltonian is frustrated.

---

[9]In previous sections, a sign was necessary in the definition of $H_{\text{pivot}}^{(1)}$. However, the sign can be absorbed into the definition of $(-1)^{\Delta_{abc}}$ and can therefore be dropped

We now derive the Hamiltonian obtained by pivoting the trivial Hamiltonian $H_0 = -\sum_v X_v$ by the 2D pivot. Note that

$$U = e^{-i\pi H_{\text{pivot}}^{(2)}} = \prod_{\Delta_{abc}} (-1)^{s_a s_b s_c} = \prod_{\Delta_{abc}} CCZ_{abc} \tag{27}$$

where $CCZ$ is known as the Controlled-controlled-$Z$ gate. Thus, evolving $H_0$ with the above unitary, we obtain the following SPT Hamiltonian on the triangular lattice:

$$H_{\text{SPT}} = -\sum_v \quad X_v \tag{28}$$

where each dense edge connecting two vertices denotes a Controlled-$Z$ gate. The decorated domain-wall picture is also evident from the form of this Hamiltonian: for each local term on the $A$ sublattice, $X_v$ creates a domain wall and the product of controlled-$Z$ operators surrounding it creates a 1D SPT long this domain wall (which naturally lives on the $B$ and $C$ sublattices).

Before proceeding, it is helpful to clarify all the possible symmetries that protect this SPT phase. Firstly, it can be protected by a $\mathbb{Z}_2^3$ symmetry generated by flipping the spins on each sublattice individually:

$$P_A = \prod_{v \in A} X_v, \qquad P_B = \prod_{v \in B} X_v, \qquad P_C = \prod_{v \in C} X_v. \tag{29}$$

Furthermore, the SPT is still protected even if we restrict to the diagonal $\mathbb{Z}_2$ symmetry $P = P_A P_B P_C$, in which case, it is in the same phase as the Levin-Gu SPT [12]. Lastly, if we restrict to the following $\mathbb{Z}_2^2$ subgroup generated by $P_A P_B$ and $P_B P_C$ (i.e. flipping the spins on two of the three colors at a time), then the SPT remains non-trivial as long as we also include time-reversal, acting as complex conjugation. To conclude, the Hamiltonian is protected by either $\mathbb{Z}_2$, $\mathbb{Z}_2^2 \times \mathbb{Z}_2^T$, or $\mathbb{Z}_2^3$ symmetries as defined above.

## 4.3 SPT interpolation

The fact that the pivot Hamiltonian generates the SPT has been pointed out in Ref. [71]. Here we show moreover that $H_{\text{pivot}}$ commutes with $H_0 + H_{\text{SPT}}$. This follows from our general theorem in Section 2.2, since $6 = k < N = 8$. However, we can also show the $U(1)$ symmetry explicitly, similarly to the 1D case. First, for convenience, we define the following "ring" operator consisting of the product of six Controlled-$Z$ operators around each vertex:

$$\mathcal{O}_v^{\text{ring}} = \tag{30}$$

Then we see that

$$H_0 + H_{\text{SPT}} = -\sum_v 2 X_v \mathcal{P}_v \tag{31}$$

where $\mathcal{P}_v$ is the projector to the subspace where $\mathcal{O}_v^{\text{ring}} = 1$

$$\mathcal{P}_v = \frac{1 + \mathcal{O}_v^{\text{ring}}}{2}. \tag{32}$$

It now suffices to show that $[X_v \mathcal{P}_v, H_{\text{pivot}}] = 0$ for any vertex. Denote the six sites around the vertex $v$ in order as $\{1, 2, 3, 4, 5, 6\}$, then we see that

$$\begin{aligned}
[X_v \mathcal{P}_v, H_{\text{pivot}}] &= [X_v, H_{\text{pivot}}]\mathcal{P}_v \\
&= (Z_1 Z_2 + Z_3 Z_4 + Z_5 Z_6 - Z_2 Z_3 - Z_4 Z_5 - Z_6 Z_1)\mathcal{P}_v
\end{aligned} \tag{33}$$

where in the first line, we used the fact that $\mathcal{P}_v$ and $H_{\text{pivot}}$ commute since they are both diagonal, and in the second line, we evaluated the commutator, which requires only the six triangles in $H_{\text{pivot}}$ that contains $v$. Lastly, one can verify that the remaining expression is annihilated by the projector, proving our claim.

Although this direct interpolation has a nice $U(1)$ pivot symmetry, it does not give a direct SPT transition. Recent work in Ref. [48] found an intermediate ferromagnetic (FM) phase where the $G_U = \mathbb{Z}_2^3$ symmetry is spontaneously broken. That is, each sublattice hosts two degenerate states, corresponding to the $\mathbb{Z}_2^3$ FM. This can be seen as the analogue of our 1D phase diagram in Fig. 3, where interpolating between the trivial and cluster SPT phase for negative values of $J$ resulted in an intermediate $\mathbb{Z}_2^2$ FM phase. This suggests that adding a same-sublattice Ising coupling to the triangular lattice—which preserves the $U(1)$ pivot symmetry—could be used to drive the system to an interesting multicritical point, similar to our 1D phase diagram in Fig. 3. Since this requires state-of-the-art Monte Carlo simulations, it goes beyond the scope of the present work. We study the resulting phase diagram in a companion work [50], where we indeed find a phase diagram similar to Fig. 3, in this case with a continuous multicritical point described by deconfined quantum criticality.

We have also considered other lattices. E.g., one can repeat the same exercise for the Union Jack lattice, which is also three-colorable. While the construction of the pivot goes through, we do not find a $U(1)$ pivot symmetry for $H_0 + H_{\text{SPT}}$ in this case. This is consistent with the pivot still having $N = 8$, but now also $k = 8$ due to the increased connectivity of the Union Jack lattice. Similarly, we have checked various 3D lattices and were not able to determine a $U(1)$ pivot symmetry. As a concrete example, the BCC lattice has $N = 16$ and $k = 24$, violating our criterion.

## 5 Pivoting with the toric code: $(d-1)$-form SPT

In this section, we consider a topological order (the toric code) as the pivot Hamiltonian. The resulting SPT phase will be protected by a combination of time-reversal and a $\mathbb{Z}_2$ $(d-1)$-form symmetry (referred to as $B^{d-1}\mathbb{Z}_2$)[10]. Alternatively, if we stay in the constraint Hilbert space of closed loop configurations (i.e. a $\mathbb{Z}_2$ gauge theory), then the SPT model can be considered as a distinct confined phase from the trivial one in the presence of time-reversal. In the main text, we discuss in detail the 2D case on a square lattice, and comment on generalizations to hypercubic lattices in higher dimension. In the appendix, we discuss a general construction on an arbitrary (dual) lattice.

We place qubits on each edge of the square lattice. The product state Hamiltonian is given by

$$H_0 = -\sum_e X_e. \tag{34}$$

We consider the toric code to be our pivot

---

[10]The response to a $d$-form gauge field $B$ of this SPT is given by $w_1 B$, where $w_1$ is the first Stiefel-Whitney class.

$$H_{\text{toric}} = -\sum_p \begin{matrix} & Z & \\ Z & & Z \\ & Z & \end{matrix} \;-\; \sum_v \begin{matrix} & X & \\ X & & X \\ & X & \end{matrix} \tag{35}$$

$$H_{\text{pivot}} = \frac{1}{4} H_{\text{toric}} \tag{36}$$

As before, the normalization $1/4$ is determined such that $e^{-2\pi i H_{\text{pivot}}} = 1$.

Both Hamiltonians are real and commute with a 1-form symmetry, defined as a product of $X$ operators around an arbitrary closed loop of the dual lattice (note that the vertex term of the toric code is itself one such loop). If the toric code vertex terms were imposed as a gauge constraint, then the ground states consist of closed loop configurations, where a loop corresponds to an eigenstate $-1$ of the $X$ operator. The gauge constraint corresponds to setting the vertex term in Eq. (35) to unity, in which case one does not need to include it in the pivot; indeed, our phase diagrams will not depend on its presence.

## 5.1   $\mathbb{Z}_2^T$ SPT in constrained Hilbert space

The result of the evolution by the pivot gives the following Hamiltonian.

$$H_{\text{SPT}} = \sum_e \left[ X_e \prod_{e' \in n(e)} Z_{e'} \right] = \sum_e \begin{matrix} & Z & & Z & \\ Z & & X & & Z \\ & Z & & Z & \end{matrix} \tag{37}$$

Here, $n(e)$ denotes the set of neighbors of each edge $e$, defined as those that are both a boundary of a common plaquette. We remark that similarly to 1D, we can put a global minus sign in front of $H_{\text{SPT}}$ at the cost of making the toric code Hamiltonian staggered. In that case, the ground state of $H_{\text{SPT}}$ is a cluster state, whose graph is given by connecting all neighboring edges on the lattice.

The ground state SPT is protected by time-reversal $\mathcal{T} = K$, and the 1-form symmetry. To see that this is indeed non-trivial, we note that due to usual arguments of symmetry fractionalization [5, 72], in a symmetric gapped phase, the string operator defining the 1-form symmetry must have long-range order for a suitable choice of endpoint operator. In the trivial phase $H_0$, this is clearly given by simply terminating the string, i.e., the ground state of $H_0$ has long-range order in an open string on the dual lattice

$$X \quad X \quad \cdots \quad X \quad X \tag{38}$$

By conjugating this by the above SPT-entangler (or alternatively, creating a string by multiplying a product of local terms appearing in $H_{\text{SPT}}$), we see that the ground state of $H_{\text{SPT}}$ has long-range order in the string

$$\begin{matrix} & Z & & & & & Z & \\ Z & & Y & X & \cdots & X & Y & & Z \\ & Z & & & & & Z & \end{matrix} \tag{39}$$

In particular, we see that two SPT phases are distinguished by whether the endpoint operator of the above string operators is even or odd under $\mathbb{Z}_2^T$.

## 5.2 Direct interpolation between SPT phases: $U(1)$ pivot superfluid

Let us now study the possible transitions between these SPT phases. As before, we start by considering the direct interpolation:

$$H = (1 - \alpha)H_0 + \alpha H_{\text{SPT}}. \tag{40}$$

It turns out that the halfway point ($\alpha = 1/2$) has $H_{\text{pivot}}$ as a $U(1)$ symmetry. Moreover, this is spontaneously broken in the ground state, making the $\alpha = 1/2$ point an infinitely-degenerate first-order transition between the two distinct SPT phases. Both of these properties can most easily be read off by gauging the 1-form symmetry of the model. A non-local (Kramers-Wannier type) mapping can be defined a la Wegner [73], which maps to qubits living on the plaquettes as follows

$$\prod_{e \subset p} Z_e \to Z_p$$

$$X_e \to \prod_{p \supset e} X_p \tag{41}$$

It is convenient to go to the dual square lattice where plaquettes are now vertices. In this lattice, we see that

$$H_0 \to -\sum_{\langle vv' \rangle} X_v X_{v'} \tag{42}$$

$$H_{\text{pivot}} \to -\frac{1}{4} \sum_v Z_v \tag{43}$$

$$H_{\text{SPT}} \to -\sum_{\langle vv' \rangle} Y_v Y_{v'}. \tag{44}$$

Hence, the above interpolation maps to

$$\tilde{H} = -\frac{1}{2} \sum_{\langle vv' \rangle} (X_v X_{v'} + Y_v Y_{v'})$$

$$- \left(\frac{1}{2} - \alpha\right) \sum_{\langle vv' \rangle} (X_v X_{v'} - Y_v Y_{v'}). \tag{45}$$

which is just the XY model on the dual square lattice at $\alpha = 1/2$. It is thus clear that $\tilde{H}_{\text{pivot}}$ commutes with $\tilde{H}$ exactly at this value of $\alpha$. Moreover, it is well-known that the ground state of the XY model is a superfluid [74]. Detuning $\alpha$ away from $1/2$ explicitly breaks the $U(1)$ symmetry down to the $\mathbb{Z}_2$ symmetry generated by $\prod Z$; note that this symmetry needs to be gauged to return to the original model in Eq. (40), such that we recover the two gapped symmetry-preserving phases.

This (infinitely-degenerate) first order transition can be seen as the two-dimensional analogue of the $c = 1$ criticality which we saw in the one-dimensional context in Sec. 3.2. Indeed, the Mermin-Wagner-Hohenberg theorem forbids a superfluid in one dimension, replacing it with quasi-long-range order. To find a direct *continuous* transition in this two-dimensional setting, we need an extra tuning parameter. We explore two different options, both giving rise to a continuous multicritical point, analogous to the one-dimensional cases explored in Sec. 3.2.

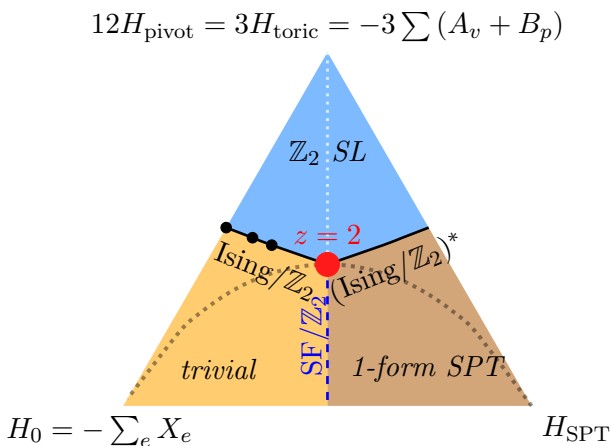

Figure 6: **BEC/$\mathbb{Z}_2$ transition as a 1-form SPT transition in 2+1 dimensions.** We use the toric code on the square lattice as a pivot to construct an SPT phase: i.e., $H_{\mathrm{SPT}} = -\sum_e U X_e U^\dagger$ with $U = \exp\left(-i\pi H_{\mathrm{toric}}/4\right) = \exp\left(-i\pi H_{\mathrm{pivot}}\right)$. The central vertical axis has a $U(1)$ symmetry generated by the toric code itself. In particular, tuning by $H_{\mathrm{toric}}$ leads to a BEC transition of the $m$-anyon; correspondingly, the red dot is a direct SPT transition described by a gauged version of the dilute Bose gas BEC/$\mathbb{Z}_2$, which we denote by $z_{\mathrm{dyn}} = 2$. The rescaling factor of 12 is included to improve presentation in barycentric coordinates, such that the $2 + 1D$ Ising transitions at the edge of the triangle are now roughly halfway the edges. The three black dots for the Ising transition were obtained in Ref. [75] for the dual XY model. The white dotted line has the toric code state as its ground state; this also holds for the $z = 2$ multicritical point. The black dotted line gives a frustration-free path where the ground state admits an exact PEPS representation with virtual bond dimension $D = 2$. More precisely, writing $H = aH_0 + b\left(3H_{\mathrm{toric}}\right) + cH_{\mathrm{SPT}}$, the solvable line is the one-parameter family $c = 1 - a$ and $b = \frac{4}{3}\sqrt{a(1 - a)}$. This includes the BEC/$\mathbb{Z}_2$ transition at $a = c = \frac{1}{2}$ and $b = \frac{2}{3}$, where the ground state is given by the fixed-point toric code wavefunction. The two Ising* transitions are distinct symmetry-enriched versions.

## 5.3    BEC/$\mathbb{Z}_2$: SPT multicriticality with $z_{\mathrm{dyn}} = 2$

Our first way of generalizing the direct interpolation in Eq. (40) is by adding the pivot itself as a tuning parameter:

$$
\begin{aligned}
H &= (1 - \alpha)H_0 + \alpha H_{\mathrm{SPT}} + hH_{\mathrm{pivot}} \\
&= (1 - \alpha)H_0 + \alpha H_{\mathrm{SPT}} + \frac{h}{4}H_{\mathrm{toric}}.
\end{aligned}
\tag{46}
$$

Note that since $H_{\mathrm{pivot}}$ commutes with both the 1-form symmetry and $\mathbb{Z}_2^T$ symmetry, the SPT phases remain distinct phases of matter. Under the aforementioned Kramers-Wannier duality (41), this maps to

$$
\tilde{H} = -\sum_{\langle vv'\rangle}\left[(1 - \alpha)X_v X_{v'} + \alpha Y_v Y_{v'}\right] - \frac{h}{4}\sum_v Z_v.
\tag{47}
$$

Its phase diagram has been explored before on the square lattice [75], which we reproduce in the dual SPT formulation in Fig. 6. We note that although various phase diagrams

of the toric code have been studied [76–81], the transition to the SPT phase in (46) has not. Clearly for large $h$, Eq. (46) will be in the toric code phase. This gapped phase is separated from the small-$h$ phase (where the $U(1)$ pivot is spontaneously broken) by a BEC transition where the $m$ anyon condenses. This is a gauged version of the usual BEC transition, and hence we denote it as BEC/$\mathbb{Z}_2$. We see that in Fig. 6, this BEC/$\mathbb{Z}_2$ serves as a continuous SPT transition if we slightly perturb in the horizontal direction. Note that the BEC transition is described by the dilute Bose gas at its upper critical dimension [82–84] and thus has dynamic critical exponent $z_{\text{dyn}} = 2$.

At the multicritical point, the ground state of Eq. (46) is exactly solvable and turns out to correspond to the fixed-point toric code wavefunction! This can be seen most easily by proving its dual statement, namely that at the multicritical point of Eq. (47) the ground state is a product state $\prod_v |\uparrow\rangle_v$. To prove this, note that up to a global constant, for $\alpha = 1/2$ and $h = 8$ we can write Eq. (47) as

$$\tilde{H} = \sum_{\langle v,v' \rangle} \Gamma_{v,v'}^\dagger \Gamma_{v,v'} \tag{48}$$

$$\Gamma_{v,v'} = \sigma_v^\dagger - \sigma_{v'}^\dagger \tag{49}$$

Observe that this has a positive spectrum, and the aforementioned product state is annihilated by $\Gamma_v$, making it a ground state of $\tilde{H}$. (In fact, one can similarly argue that it is a ground state for the white dotted line in Fig. 6.) Despite the ground state of Eq. (46) being in the fixed-point toric code state, the Hamiltonian is gapless: $m$ anyons have a quadratic dispersion $\varepsilon_k \sim k^2$. Although a (different) gapless 'uncle' Hamiltonian for the toric code wavefunction has been discussed before [85, 86], we are not aware of any study of its nearby phase diagram or its connection to SPT physics.

There is also an exactly-solvable line which tunes through $0 \leq \alpha \leq 1$. Indeed, the continuous SPT multicritical point lies on a 1-parameter family of frustration-free models. This path is given by $h = 4z_{\text{co}} \sqrt{\alpha(1-\alpha)}$, where $z_{\text{co}} = 4$ is the coordination number of the square lattice. While this has been noted before (for the dual XY model) on a square lattice [75], it can be readily proven for arbitrary lattices (see Appendix D where we use the Witten conjugation method [87, 88]). This path is represented in Fig. 6 as a gray dotted line. Along this line, the ground state can be written as an exact projected pair entangled state (PEPS) [89] with virtual bond dimension $D = 2$. More concretely, for the part of the path lying within the trivial phase (i.e., $\alpha < 1/2$), the ground state is obtained by performing an imaginary time evolution with the toric code Hamiltonian on the ground state of $H_0$

$$\exp\left(-\beta H_{\text{toric}}\right) \prod_e |\rightarrow\rangle_e \tag{50}$$

where $\beta = \frac{1}{4} \text{arsech}(1 - 2\alpha)$ (see derivation in Appendix D). Notably, we see that $\alpha = 1/2$ corresponds to the limit $\beta \to \infty$. This gives another way of seeing that at the multicritical point, the ground state coincides with that of $H_{\text{toric}}$.

We note that these exactly-solvable ground states that tune through the three distinct phases in Fig. 6 and which are connected at a multicritical point, constitute a two-dimensional generalization of the concept of the 'tensor network skeleton' introduced recently in Ref. [63] (which studied it in the one-dimensional setting with matrix product states).

## 5.4   $O(2)/\mathbb{Z}_2$ multicriticality and emergent $U(1)$ pivot symmetry

The same two 1-form SPT phases can be separated by a multicritical point described by $O(2)/\mathbb{Z}_2$ criticality. In contrast to the BEC/$\mathbb{Z}_2$ transition above, this is a 2+1d CFT, with

dynamical critical exponent $z = 1$. It is obtained from the $O(2)$ Wilson-Fisher fixed point by gauging the $\mathbb{Z}_2$ subgroup of the rotation symmetry. The $\mathbb{Z}_2$ entangler which swaps the 1-form SPT phases discussed above can be identified with the $\mathbb{Z}_4 \subset O(2)$ subgroup before gauging; the $O(2)/\mathbb{Z}_2$ criticality has a full $U(1)$ symmetry which can thus be interpreted as the pivot symmetry. The 1-form symmetry is the magnetic symmetry generated by the $\mathbb{Z}_2$ Wilson line. The mixed anomaly occurs because the entangler squares to a gauge transformation.

A nice feature of the $O(2)/\mathbb{Z}_2$ critical point is that the $U(1)$ pivot symmetry is emergent just assuming the $\mathbb{Z}_2$ entangler. Indeed, in the $O(2)$ CFT, among rotation-charged $O(2)$ operators, only those of charges $< 4$ are RG-relevant [90, 91]. When we gauge the $\mathbb{Z}_2$ subgroup of rotations to obtain the $O(2)/\mathbb{Z}_2$ CFT, we project out the odd charged operators and those with charge 2 mod 4 are further charged under the entangler.

The nearby phase diagram of this critical point can be derived by gauging the nearby phase diagram of the usual $O(2)$ point. If we preserve the $\mathbb{Z}_2$ entangler, there is a single relevant direction, given by the mass term $m^2|\varphi|^2$ of the $O(2)$ field, which tunes between a toric code phase (for positive $m^2$) and a spontaneous-entangler breaking phases (for negative $m^2$).

If we break the $\mathbb{Z}_2$ entangler, there are two relevant operators Re $\varphi^2$ and Im $\varphi^2$. If we include a time reversal symmetry acting by complex conjugation $\varphi \mapsto \varphi^*$, only Re $\varphi^2$ is allowed. With positive coefficient and $m^2 < 0$, the Higgs condensate is imaginary and we get an SPT protected by time reversal and the 1-form symmetry. For the other sign, we get a real Higgs condensate and a trivial phase. This critical point thus has the same nearby phases as the BEC/$\mathbb{Z}_2$ transition shown in Fig. 6.

As a possible lattice realization of the above field theory discussion, we can simply modify the Hamiltonian in Sec. 5.3 by staggering the plaquette term $B_p$. Indeed, when we ungauge the $\mathbb{Z}_2$ gauge symmetry, this amounts to perturbing the $XY$ model with a staggered field $\sum_v (-1)^v Z_v$, rather than the homogeneous field in Eq. (47). The result is that we will stay in the zero-spin sector, rather than tuning us to the maximally polarized state. It is thus natural to expect that tuning this staggered field will drive an $O(2)$ transition into the paramagnetic phase, with the original model thus realizing the $O(2)/\mathbb{Z}_2$ criticality discussed above. It would be interesting to numerically study this in future work.

## 5.5 Construction in higher dimensions

We consider the model on a hypercubic lattice in $d$ dimensions. (In general, we can choose any Voronoi cellulation, i.e., the dual of some general lattice. The model and the construction of the exactly path of Hamiltonians is considered in Appendix D) The degrees of freedoms live on the codimension-1 faces of the lattice. The trivial Hamiltonian is given by

$$H_0 = -\sum_f X_f \tag{51}$$

The pivot is the toric code Hamiltonian in $d$ dimensions

$$H_{\text{pivot}} = \frac{1}{4} H_{TC} \tag{52}$$

$$H_{TC} = -\sum_{f \subset c} Z_f - \sum_{f \supset e} X_e \tag{53}$$

where $e$ are codimension-2 edges, and $c$ are the (top-dimensional) hypercubes. These Hamiltonians commute with a $(d-1)$-form symmetry $B^{d-1}\mathbb{Z}_2$ defined as a product of X

operators around an arbitrary closed loop of the dual lattice. Evolving $H_0$ by the pivot, we obtain

$$H_{\text{SPT}} = \sum_f \left[ X_f \prod_{f' \in n(f)} Z_{f'} \right] \tag{54}$$

where $n(f)$ denotes the set of faces which share a common boundary edge. This SPT is protected by a combination of time-reversal $\mathcal{T} = K$ and $B^{d-1}\mathbb{Z}_2$. Its non-triviality can be similarly seen by the fractionalization of an open string operator, and a Kramers-Wannier duality maps the system to the XY model on the dual hypercubic lattice. This confirms that the midpoint of the direct interpolation has a $U(1)$ pivot symmetry in all dimensions.

# 6  Outlook

In this work, we have introduced the notion of a pivot Hamiltonian as continuous generators of SPT entanglers. These Hamiltonians can then naturally play two roles: one as a generator of entanglement, the other as a symmetry generator at SPT transitions. The former has been explicitly demonstrated by showing how using the Ising and cluster models as pivots generate a whole web of dualities. The latter role has been encountered in the various models of using Ising, staggered Baxter-Wu and toric code Hamiltonian as pivots where we confirmed a $U(1)$ pivot symmetry in the interpolated model.

The aforementioned 'duality web' of 1d models naturally lies along a line. In higher dimensions it may be interesting to explore similar structures. It appears from some preliminary exploration that the combinatorial structure of the web is more general.

So far we have focused on $\mathbb{Z}_2$ entanglers acting as symmetries of SPT transitions. More generally we can consider multicritical points where $n$ SPT phases that are cyclically related by a $\mathbb{Z}_n$ entangler meet. Because $U(1)$ has only a $\mathbb{Z}_2$ class of automorphisms, the algebra of the protecting symmetry $G$ and the $U(1)$ pivot has to be larger than any semi-direct product $U(1) \rtimes G$. This suggests there may be interesting critical points with symmetry enhancement in phase diagrams where a $\mathbb{Z}_n$ orbit of SPTs meet.

There is a general method of constructing SPT entanglers from group cocycles. In particular, we can express SPT ground states as paramagnet states dressed with certain phase factors [53]. These phase factors define a diagonal operator whose logarithm gives a $U(1)$ pivot. However, this pivot has certain ambiguities, and may not be amenable to constructing a $U(1)$-symmetric SPT transition. Can we improve the general construction of entanglers to have this nicer property?

For continuous $G$ and fermionic systems, a general construction of entanglers is lacking. For example, can we realize an $SO(3) \times U(1)$ Haldane SPT transition on the lattice with on-site $SO(3)$ and pivot $U(1)$?

We have observed that pivots give rise to generalized Thouless pumps, and the pivot becomes a symmetry of the diabolical point. However, there are pumping families which are not associated with SPT transitions. For example in the 1+1D $\mathbb{Z}_2^2$ cluster example, the family is protected just by pumping a charge under the diagonal symmetry $\prod_n X_n$, and we can break the other protecting symmetries so that there is no SPT at $H(\pi)$. Does this give a more general context for pivots?

Finally, in Sec. 5.3, we studied a model where the toric code ground state appears as the ground state of a gapless model. This is reminiscent of the construction of 'uncle Hamiltonians' in [85]. It may be interesting to study the nearby phase diagrams of these models.

## Acknowledgments

We thank Shu-Heng Shao for helpful discussions. NT is supported by NSERC. RV and AV are supported by the Simons Collaboration on Ultra-Quantum Matter, which is a grant from the Simons Foundation (651440, A.V.). RV is supported by the Harvard Quantum Initiative Postdoctoral Fellowship in Science and Engineering.

## A   Technical details about 1D models

### A.1   Proof of $U(1)$ pivot symmetry

Here we show that in the 1D case, $H_0 + H_{\text{SPT}}$ has a $U(1)$ pivot symmetry without relying on a non-local Kramers-Wannier transformation. To see this, first observe that

$$\frac{1}{2}(H_0 + H_{\text{SPT}}) = -\sum_n X_n \mathcal{P}_n \tag{55}$$

where $\mathcal{P}_n = \frac{1 + Z_{n-1}Z_{n+1}}{2}$ is a projector. Note that one can interpret Eq. (55) as a hopping term for domain walls, which thus commutes with $H_{\text{pivot}}$. We can also show more explicitly that $X_n \mathcal{P}_n$ commutes with $H_{\text{pivot}}$:

$$
\begin{aligned}
[X_n \mathcal{P}_n, H_{\text{pivot}}] &= [X_n, H_{\text{pivot}}]\mathcal{P}_n \\
&= (-1)^n [X_n, Z_n Z_{n+1} - Z_{n-1} Z_n]\mathcal{P}_n \\
&\propto [X_n, Z_n]\underbrace{(Z_{n+1} - Z_{n-1})\mathcal{P}_n}_{=0}.
\end{aligned}
\tag{56}
$$

In the last step we used that $\mathcal{P}_n$ is a projector onto states that satisfy $Z_{n-1} = Z_{n+1}$.

### A.2   Field theory of the SPT transition

Here we give the field theory describing the critical point in Eq. (13). Let $\varphi(x)$ and $\theta(x)$ denote two conjugate $2\pi$-periodic fields (i.e. $[\partial_x \theta(x), \varphi(y)] = 2\pi i \delta(x - y)$), then the low-energy theory at $\alpha = 1/2$ is described by

$$H_{LL} = \frac{1}{2\pi}\int\left(K(\partial_x \varphi)^2 + \frac{1}{4K}(\partial_x \theta)^2\right)\mathrm{d}x. \tag{57}$$

Here $K$ is the Luttinger liquid parameter which labels the one-parameter family of compact boson CFTs; equivalently, one sometimes speaks of the compactification radius $r_c = \sqrt{K}$ [92]. This labels the scaling dimensions: $[e^{i(n\varphi + m\theta)}] = \frac{n^2}{4K} + m^2 K$. In our case, we are at one of the two free-fermion values; in particular, if we take the usual convention that the $U(1)$ symmetry is generated by $\partial_x \theta$, then $K = 1/4$. The dictionary relating the lattice operators and these effective low-energy field operators is as follows (where we suppress unknown numerical prefactors):

$$(-1)^n Z_n Z_{n+1} \sim \partial_x \theta + (-1)^n \sin(2\theta) \tag{58}$$

$$Z_{n-1} Z_{n+1} \sim (\partial_x \theta)^2 \tag{59}$$

$$X_n - Z_{n-1}X_n Z_{n+1} \sim \cos\varphi. \tag{60}$$

Relatedly, the symmetries of interest act as

$$U = e^{-i\alpha H_{\text{pivot}}} : \quad \varphi \to \varphi + \alpha, \quad \theta \to \theta \tag{61}$$

$$P : \quad \varphi \to \varphi, \quad \theta \to \theta + \pi \tag{62}$$

$$P_1 : \quad \varphi \to -\varphi, \quad \theta \to -\theta \tag{63}$$

$$P_2 : \quad \varphi \to -\varphi, \quad \theta \to \pi - \theta \tag{64}$$

$$\mathcal{T} : \quad \varphi \to -\varphi, \quad \theta \to \theta \tag{65}$$

$$\text{translation} : \quad \varphi \to -\varphi, \quad \theta \to \frac{\pi}{2} - \theta. \tag{66}$$

This theory has a chiral anomaly which matches the general form of an SPT transition described in Sec. 2. For example, restricting to the group generated by the entangler (coupling to a $\mathbb{Z}_2$ gauge field $A$), $P$ (coupling to a $\mathbb{Z}_2$ gauge field $B$), and $P_1$ (coupling to a $\mathbb{Z}_2$ gauge field $C$), we find the anomaly $\frac{1}{2}ABC$, indicating that the pivot creates the $\mathbb{Z}_2 \times \mathbb{Z}_2$ cluster SPT with $\omega = BC$.

## A.3   KT multicritical point

Here we study the phase diagram of Eq. (15), as shown in Fig. 3.

Let us start with the $c = 1$ criticality at $\alpha = 1/2$ with $J = 0$. We have already mentioned that this corresponds to the free-fermion point with $K = 1/4$. Due to the explicit $U(1)$ pivot symmetry, we know that no $\cos(m\varphi)$ or $\sin(m\varphi)$ perturbation can be generated at low energies. Moreover, the spin-flip symmetry $P$ forbids $\cos(\theta)$ and $\sin(\theta)$. In fact, even $\cos(2\theta)$ is forbidden if we keep translation symmetry (see Eq. (66)) and $\sin(2\theta)$ would violate the sublattice symmetry $P_1$ (see Eq. (62)). Hence, the dominant symmetry-allowed perturbation is $\cos(4\theta)$, which has scaling dimension $16K$. For $J = 0$ (where $K = 1/4$) this has dimension 4, making it irrelevant. However, tuning $J \neq 0$ introduces the marginal operator (see Eq. (59)). Eventually, $K \to 1/8$, at which point the $\cos(4\theta)$ perturbation becomes marginal. Beyond this point, we expect a symmetry-breaking phase. Indeed, in the phase diagram in Fig. 3, we find a fourfold degenerate phase for large negative $J$, consistent with the limit $J \to -\infty$ where have a simple Ising ferromagnet on each of the two sublattices. (In our 2D example in Sec. 4, this phase will be replaced by an 8-fold degenerate ferromagnet on each of the three sublattices of the triangular lattice.) In this case, we find that these two regimes are separated by Kosterlitz-Thouless (KT) criticality. (In 2D, we will find an exotic $SO(5)$ deconfined critical point.)

As we tune $\alpha \neq 1/2$, we no longer have a $U(1)$ pivot symmetry. This means there is no longer a mutual anomaly with the symmetries protecting the SPT phase, such that we can have symmetric gapped phases. Indeed, in Fig. 3 we find the two distinct SPT phases, which are separated from the fourfold degenerate ferromagnet by Ashkin-Teller criticality. The latter can intuitively be thought of as an Ising criticality on each of the two sublattices (these are marginally coupled by an energy-energy coupling which allows for them to smoothly connect to the compact boson line, with the KT point serving as a juncture between the two [92]).

## A.4   Kramers-Wannier duality: making the pivot local

Here we briefly review how the 1D Hamiltonians encountered in the main text far can be mapped to more conventional ones. While this requires a non-local mapping which can obscure the physics at play, it can be useful to obtain phase diagram and/or relate it to known physics.

Let us consider the following (non-local) Kramers-Wannier transformation:

$$X_n \to -X_{n-1}X_n \qquad\qquad (-1)^n Z_n Z_{n+1} \to Z_n. \qquad (67)$$

If we ignore boundary issues (e.g., let us consider an infinitely long chain), then these new operators satisfy the desired Pauli algebra. This can be thought of as gauging the global $\mathbb{Z}_2$ symmetry. Starting with the Hamiltonians $H_0, H_{\text{SPT}}, H_{\text{NNN}}, H_{\text{pivot}}$ defined in Sec. 3, we denote the resulting Hamiltonians after this mapping with tildes as follows:

$$\tilde{H}_0 = \sum_n X_n X_{n+1}, \qquad (68)$$

$$\tilde{H}_{\text{SPT}} = \sum_n Y_n Y_{n+1}, \qquad (69)$$

$$\tilde{H}_{\text{NNN}} = -\sum_n Z_n Z_{n+1}, \qquad (70)$$

$$\tilde{H}_{\text{pivot}} = \sum_n Z_n. \qquad (71)$$

Thus, we see that Eq. (15) maps exactly to the XXZ chain, and the pivot maps to the chemical potential term for the bosons. In particular, the direct interpolation in Eq. (13) maps to the XX chain (making the $U(1)$ pivot symmetry manifest), whereas the KT transition (red dot in Fig. 3) is dual to the antiferromagnetic spin-1/2 Heisenberg chain. Indeed, this mapping elucidates how Fig. 3 can be directly obtained from the well-known phase diagram of the XYZ chain.

We note that for the Kramers-Wannier duality in Fig. 4, we need to perform Eq. (67) *and* a Hadamard transformation, i.e., swapping $X \leftrightarrow Z$.

# B  Sufficient condition for enhanced $U(1)$ symmetry

We describe a sufficient condition for the $\mathbb{Z}_2$ symmetry at $\alpha = 0.5$ of Eq. (4) to be enhanced to a full $U(1)$ symmetry for pivots that are diagonal. Interestingly, this turns out to be a geometric constraint. Define a bipartite graph consisting of a set of vertices $V = \{V_0 \cup \tilde{V}\}$ and edges $E$ connecting vertices from $V_0$ to $\tilde{V}$. We take the Hilbert space to be a tensor product of qubits, each living on some vertex $v \in V_0$. The trivial Hamiltonian is

$$H_0 = -\sum_{v \in V_0} X_v \qquad (72)$$

Furthermore, the pivot is a sum of local terms, each positioned at vertices $\tilde{v} \in \tilde{V}$.

$$H_{\text{pivot}} = \frac{1}{N} \sum_{\tilde{v} \in \tilde{V}} H_{\tilde{v}}^Z \qquad (73)$$

$$H_{\tilde{v}}^Z = \pm \prod_{(v\tilde{v}) \in E} Z_v \qquad (74)$$

where $N$ is the largest integer which properly normalizes the pivot $e^{2\pi i H_{\text{pivot}}} = 1$. By definition, $X_v$ and $H_{\tilde{v}}^Z$ anticommute if $(v\tilde{v}) \in E$.

The $U(1)$ symmetry is most manifest by performing an isomorphism at the level of operators, to a dual Hilbert space where qubits are instead placed on $\tilde{V}$. The map is given

by

$$X_v \to \prod_{(v\tilde{v}) \in E} X_{\tilde{v}}, \tag{75}$$

$$H_{\tilde{v}}^Z \to Z_{\tilde{v}}. \tag{76}$$

In this basis, it is then clear that evolution by the pivot is just a rotation around the $Z$-axis for all qubits in $\tilde{V}$.

There are various incarnations of this isomorphism [93–101]. It is often called the generalized Kramers-Wannier duality, or the gauging map in quantum information theory. (Alternatively, it can be obtained by performing a minimal coupling $H_{\tilde{v}}^Z$ with gauge fields defined on $\tilde{V}$ and going to the effective Hilbert space where the Gauss law located at each $v \in V_0$ is enforced.) In the dual Hilbert space, we see that

$$\tilde{H}_0 = -\sum_{v \in V_0} \prod_{(v\tilde{v}) \in E} X_{\tilde{v}}, \tag{77}$$

$$\tilde{H}_{\text{pivot}} = \frac{1}{N} \sum_{\tilde{v} \in \tilde{V}} Z_{\tilde{v}}. \tag{78}$$

Note that depending on the graph, it is possible for there to be constraints on the dual Hilbert space. However, we can still argue the presence or absence of the $U(1)$ pivot in the unrestricted Hilbert space.

The dual of the SPT can be obtained by evolving $\tilde{H}_0$ using $H_{\text{pivot}}$. We see that

$$\tilde{H}_{\text{SPT}} = -\sum_{v \in V_0} \prod_{(v\tilde{v}) \in E} \tilde{X}_{\tilde{v}} \tag{79}$$

where $\tilde{X}_{\tilde{v}}$ is just the Pauli-$X$ operator rotated by an angle $\pi/N$

$$\tilde{X}_{\tilde{v}} = e^{-i\pi Z/N} X_{\tilde{v}} e^{i\pi Z/N} \tag{80}$$

Now consider $\tilde{H}_0 + \tilde{H}_{\text{SPT}}$. The key fact we will use is that this Hamiltonian is $k$-local, where $k$ is the largest coordination number of the vertices $v \in V$. Note that a $k$-local term of qubits can at most have charge $k$ (e.g., $\sigma_1^+ \sigma_2^+ \cdots \sigma_k^+$). However, we know that the Hamiltonian must commute with

$$\exp\left(i\pi \tilde{H}_{\text{pivot}}\right) = \exp\left(i\frac{2\pi}{N} \sum_{\tilde{v} \in \tilde{V}} \frac{Z_{\tilde{v}}}{2}\right). \tag{81}$$

I.e., we are guaranteed $\mathbb{Z}_N$ symmetry. We must thus only exclude the possibility of terms which are *neutral* under $\mathbb{Z}_N$ but *charged* under the full $U(1)$. Clearly such terms would have a charge which is a multiple of $N$. Since $z$ is the largest charge we can write down, we are guaranteed $U(1)$ symmetry if $N > k$.

## C  Pivots from decorated domain walls

We generalize the decorated domain wall construction in 4.1 to arbitrary dimension. Given a pivot Hamiltonian in $d$ spatial dimensions $H_{\text{pivot}}^d$ written as a sum of local commuting terms

$$H_{\text{pivot}}^d = \sum H_{\text{pivot,loc}}^d \tag{82}$$

where $H^d_{\mathrm{pivot},loc}$ acts on $d$- dimensional simplices. If the pivot creates a $G$-SPT in $d$ dimensions, we can use this to construct a pivot in $d+1$ dimensions

$$H^{(d+1)}_{\mathrm{pivot}} = \sum_{\triangle} \frac{1}{2}(-1)^{\triangle} Z_A H^{(d)}_{\mathrm{pivot},loc} \tag{83}$$

which create a $\mathbb{Z}_2 \times G$ SPT in $d+1$ dimensions. This pivot realizes a decorated domain wall construction.

To see this, we consider the evolution of $H^{(d+1)}_{\mathrm{pivot}}$ for time $\pi$:

$$e^{\pi i H^{(d+1)}_{\mathrm{pivot}}} = \prod_{\triangle} e^{\pi i/2(-1)^{\triangle}(1-2s_A) H^{(d)}_{\mathrm{pivot},loc}} \tag{84}$$

The first term cancels because each $d$ simplex receives contributions from two tetrahedra with opposite sign. For each edge $d$-simplex, the contribution from the two adjacent $d+1$ simplices $\Delta_{i_1 jk\dots}$ and $\Delta_{i_2 jk\dots}$ is

$$\left( e^{\pi i H^{(d)}_{\mathrm{pivot},loc}} \right)^{s_{i_1} + s_{i_2}} \tag{85}$$

which precisely creates an SPT on that $d$-simplex when there is a domain wall.

The $\mathbb{Z}_2^3$ pivot follows directly from decorating the 1D cluster state pivot. In fact we can see how the Ising model itself can arise as a decorated domain wall pivot!

Consider the $0D$ trivial Hamiltonian $H_0 = -X$ and the pivot $H_{\mathrm{pivot}} = \frac{Z}{2}$. The "$0D$ SPT" created is $H_{\mathrm{SPT}} = e^{-\pi i Z/2} H_0 e^{\pi i Z/2} = +X$. Indeed, preserving the $\mathbb{Z}_2$ symmetry $X$, the two phases are separated by a first order transition (level crossing) since $H_0 + H_{\mathrm{SPT}} = 0$.

Using the recipe for the decorated domain walls above, the resulting $1D$ pivot given by decorating the $0D$ SPT ($\mathbb{Z}_2$ charges) on domain walls in 1D is $H_{\mathrm{pivot}} = \sum (-1)^n Z_n Z_{n+1}$. This is exactly the staggered Ising interaction.

## D   Exactly-solvable path

In this appendix, we use the Witten conjugation argument [87,88] to construct an exactly solvable path which interpolates two different symmetry breaking Hamiltonians with a transition through a $z = 2$ critical point. From this, the Kramers-Wannier duality relates this to an exactly solvable path interpolating between trivial and SPT phases . Consider an arbitrary graph $\mathcal{G} = (V, E)$. For all edges $e = \langle vv' \rangle \in E$, we can consider the Hamiltonian

$$\tilde{H}_0 = \sum_e \frac{1 - X_v X_{v'}}{2} = \sum_e \Gamma_e^\dagger \Gamma_e \tag{86}$$

where $\Gamma_e = (X_v - X_{v'})/2$. This Hamiltonian respects a global symmetry $\prod_v Z_v$ and complex conjugation, but whose ground states are given by $\bigotimes_v |\rightarrow\rangle_v$ and $\bigotimes_v |\leftarrow\rangle_v$, which spontaneously break the $\mathbb{Z}_2$ symmetry.

To construct an exactly solvable path, we define the following imaginary time evolution

$$M(\beta) = \prod_v e^{\beta Z_v} \tag{87}$$

for some real parameter $\beta$. Defining $\Gamma_e(\beta) = M \Gamma_e M^{-1}$, we see that

$$\tilde{H}_\beta = \sum_e \Gamma_e(\beta)^\dagger \Gamma_e(\beta) \tag{88}$$

is a frustration free Hamiltonian with ground states $M \bigotimes_v |\rightarrow\rangle_v$ and $M \bigotimes_v |\leftarrow\rangle_v$

The expression for $\Gamma_e(\beta)$ is

$$\Gamma_e(\beta) = \frac{1}{2}[X_v e^{-2\beta Z_v} - X_{v'} e^{-2\beta Z_{v'}}] \tag{89}$$

Using this, we find that up to a constant, the Hamiltonian takes the form

$$\tilde{H}_\beta = -\frac{1}{2}\sum_e \left[\cosh^2(2\beta)X_v X_{v'} + \sinh^2(2\beta)Y_v Y_{v'}\right]$$
$$-\frac{1}{4}\sinh(4\beta)\sum_v z_v Z_v \tag{90}$$

where $z_v$ is the coordination number of the vertex $v$.

It is helpful to reparametrize this family of Hamiltonians with a new parameter $\alpha = \frac{1-\mathrm{sech}\,4\beta}{2}$. Then up to a rescaling, the Hamiltonian can be written as

$$\tilde{H}_\alpha = -\sum_e \left[(1-\alpha)X_v X_{v'} + \alpha Y_v Y_{v'}\right]$$
$$-\sqrt{\alpha(1-\alpha)}\sum_v z_v Z_v \tag{91}$$

where the Hamiltonian interpolates from $XX$ to $YY$ by varying $\alpha$ from 0 to 1. Furthermore, for $\alpha = \frac{1}{2}$, the Hamiltonian reads

$$2\tilde{H}_{\alpha=\frac{1}{2}} = -\sum_e [X_v X_{v'} + Y_v Y_{v'}] - \sum_v z_v Z_v \tag{92}$$

which is the BEC point of the XY model. Since this corresponds $\beta \to \infty$, the ground state of this Hamiltonian is $\bigotimes_v |\uparrow\rangle_v$

This exactly solvable path is known for the 1D chain (from free-fermion solution), for the 2D square lattice [75] and on the 3D cubic lattice [102]. Here, we show that the results holds for the XY model defined on any lattice.

The above is a dual description of the path discussed in Sec. 5. They are related by performing the Kramers-Wannier duality which maps

$$X_v X_{v'} \to X_e \tag{93}$$
$$Z_v \to \prod_{e \supset v} Z_e \tag{94}$$

The final Hamiltonian parametrized by $\alpha$ is therefore

$$H_\alpha = -\sum_e \left[(1-\alpha)X_e - \alpha X_e \prod_{n(e)} Z_e\right]$$
$$-\sqrt{\alpha(1-\alpha)}\sum_v z_v \prod_{e \supset v} Z_e \tag{95}$$

where $n(e)$ denotes all edges that share a boundary vertex with $e$ and this Hamiltonian lives in a constraint Hilbert space $\prod_{e \subset p} X_e = 1$. This gives us a frustration-free path from a trivial Hamiltonian to an SPT protected by complex conjugation $\mathbb{Z}_2^T$ and a $(d-1)$-form $\mathbb{Z}_2$ symmetry. Notably, as per duality, the ground state at the transition $\alpha = 1/2$ is the $d$-dimensional $\mathbb{Z}_2$ toric code.

To make the connection explicit with the square lattice construction in the main text we set the coordination number $z_v = 4$, and using

$$H_0 = -\sum_e X_e, \tag{96}$$

$$H_{\text{SPT}} = \sum_e X_e \left[ \prod_{e' \in n(e)} Z_{e'} \right], \tag{97}$$

$$H_{\text{pivot}} = -\sum_v \prod_{e \supset v} Z_e, \tag{98}$$

which are the definitions of the Hamiltonians on the dual square lattice of the main text, we obtain

$$H_\alpha = (1 - \alpha)H_0 + \alpha H_{\text{SPT}} + 4\sqrt{\alpha(1 - \alpha)}H_{\text{pivot}} \tag{99}$$

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
