# Peer review of "Pivot Hamiltonians as generators of symmetry and entanglement"

_SciPost Physics_

## Round 1 · Referee Report · Anonymous (Referee 1) · 2022-5-28

Report
The paper is well written and offers an interesting perspective on SPT phases and the connections between models exhibiting them. The method is based on an original idea and has broad applicability with examples in various dimensions. I think this paper is a interesting contribution to the field and is sufficiently relevant to warrant publication in SciPost Physics Core, upon clarifying the issues below.
Requested changes
Specific comments: 1. It is stated above Eq. (3) that $H_{pivot}$ could have a smaller symmetry group than $G$, implying that it is possible that it also has the symmetry G. It is my understanding, it is crucial that it does not have the symmetry $G$ that protects the resulting SPT, as otherwise $H(\theta)$ would be a path of $G$-symmetric Hamiltonians, along which the SPT order can not change.
-
Below equation 6, it is stated that $N$ is the "smallest integer" such that $e^{2\pi i H_{pivot}} = 1$. If this condition holds for some $N$ it will hold for $N=1$, so I believe this should say "largest integer" instead. (Additionally, in the paragraph right below, there is a broken reference to Appendix 2.2, which does not exist.)
-
(optional) Figure 2 and 6 might benefit from having axis like Figure 3, although this is not strictly necessary for their interpretation.
General comment: It is mentioned that the SPT entanglers are finite depth unitaries. There has been previous work on dualities relating SPTs to trivial phases, such as the Kennedy-Tasaki transformation for the Haldane phase and its generalizations. These dualities all have the feature that a local order operator is mapped to a non-local string order operator, which is not something that can be achieved with a finite depth unitary. I understand that by breaking the symmetry the SPT entanglers can map between different SPT phases, but a comment (or speculation) on the distinction between these two seemingly different kinds of duality mappings might be helpful. Additionally, a generic pivot Hamiltonian will not generate a constant depth unitary, so some discussion on the assumptions required for this to be the case would be useful.
We thank the referee for their comments.
-
We agree that the symmetry of the pivot is strictly smaller than $G$ if the symmetry is unitary. However, there is is a subtlety when $G$ contains an antiunitary symmetry. For example, the Ising Hamiltonian Eq.1 as a pivot commutes with the full symmetry $G=\mathbb Z_2 \times \mathbb Z_2^T$ in Eq. 12. However, the rotated Hamiltonian Eq.3 only commutes with time-reversal at $\theta=0,\pi$. We have added a footnote to clarify this point.
-
We thank the referee for pointing this out. This has been corrected. The link now correctly refers to Appendix B.
-
We have added a clarification to the captions of Figure 2 and 6 that they are plotted using barycentric coordinates. We thank the referee for this suggestion.
Reply to general comment:
It is our understanding that the Kennedy-Tasaki transformation maps between a symmetry broken phase to an SPT phase, which explains why it cannot be written as a finite depth circuit. However, it should be possible to realize the Kennedy-Tasaki transformation by augmenting the SPT entanglers with Kramers-Wannier dualities. The pivot Hamiltonians in our paper are defined to to be local, and therefore it will always generate a constant depth unitary. We are unaware of non-local versions of our pivot Hamiltonians.

Author: Nathanan Tantivasadakarn on 2022-10-15 [id 2924]
(in reply to Report 2 on 2022-06-30)We thank the referee for their careful reading and for pointing out where we can improve the work. Firstly, section 4.1 has been improved, clarifying the distinction of the local pivot Hamiltonian vs the total pivot Hamiltonian. Secondly, the presentation of Sec. 5.4 is derived based on field theory arguments. Verifying this numerically in a lattice model is left to future work. In the updated version, we have added a description of a lattice model that is likely to realize this $O(2)/\mathbb Z_2$ transition, which future work can numerically explore.

---

## Round 1 · Referee Report · Anonymous (Referee 2) · 2022-6-30

Report
In all, I view this contribution as a powerful construction to explore the phase diagrams of SPTs. The presentation is lucid and clear with some exceptions. To me, section 4 was essentially unintelligible, perhaps largely due to me lack of familiarity with Ref. [52], but if that's the case, this deviation from self-containedness came without warning. The idea of creating a 2D pivot from a 1D pivot still eludes me based on what is written. There is a H(1)_pivot, which is really "zero dimensional" if I understand correctly (or a sum is missing in its introduction), and its (j,k) dependence in the first line of (21) is suppressed. The latter defines H(2)_pivot, and the relation to H_pivot in subsequent sub-sections, e.g. Eq. (26), is not clear. Moreover, the procedure usually starts by defining an H0, see above, but this is left unclear in 4.1 and 4.2, until H0 is suddenly referenced in 4.3, without definition. I am hopeful that the presentation of this section can be improved. Similarly, in Section 5.4, is would be great if the authors could say clearly if the scenario they develop there involves conjecture, is based on details that they prefer to leave to future work, or should be self-evident from the present context (which, however, then largely escapes me). Other than that, the paper is well written and certainly represents a contribution worthy of publication in SciPost.
Note in passing: There were some typos: "as as" appears twice on p. 17, p. 18 speaks of "a suitable chose".

---

## Round 2 · List of Changes

1. Exposition of Sec. 4.1 has been improved, clarifying the distinction of the local pivot Hamiltonian vs the total pivot Hamiltonian.
2. In Sec. 5.4 a description of a lattice model that is likely to realize the $O(2)/\mathbb Z_2$ transition has been added, which can be numerically explored in future work.

---

## Editorial Decision

editorial_decision: